# Local MDI+: Local Feature Importances for Tree-Based Models

**Zhongyuan Liang**[*]                                          zhongyuan_liang@berkeley.edu
*Department of Computational Precision Health*
*UC Berkeley and UCSF*

**Zachary T. Rewolinski**[*]                                          zachrewolinski@berkeley.edu
*Department of Statistics*
*UC Berkeley*

**Abhineet Agarwal**                                          aa3797@berkeley.edu
*Department of Statistics*
*UC Berkeley*

**Tiffany M. Tang**                                          ttang4@nd.edu
*Department of Applied and Computational Mathematics and Statistics*
*University of Notre Dame*

**Bin Yu**                                          binyu@berkeley.edu
*Department of Statistics and EECS*
*UC Berkeley*

**Reviewed on OpenReview:** `https://openreview.net/forum?id=TcXidnGHpA`

## Abstract

Tree-based ensembles such as random forests remain the go-to for tabular data over deep learning models due to their prediction performance and computational efficiency. These advantages have led to their widespread deployment in high-stakes domains, where interpretability is essential for ensuring trustworthy predictions. This has motivated the development of popular local (i.e. sample-specific) feature importance (LFI) methods such as LIME and TreeSHAP. However, these approaches rely on approximations that ignore the model's internal structure and instead depend on potentially unstable perturbations. These issues are addressed in the global setting by MDI+, a global feature importance method which combines tree-based and linear feature importances by exploiting an equivalence between decision trees and least squares on a transformed node basis. However, the global MDI+ scores are not able to explain predictions when faced with heterogeneous individual characteristics. To address this gap, we propose Local MDI+ (LMDI+)[1], a novel extension of the MDI+ framework that quantifies feature importances for each particular sample. Across twelve real-world benchmark datasets, LMDI+ outperforms existing baselines at identifying instance-specific predictive features, yielding an average 10% improvement in predictive performance when using only the selected features. It further demonstrates greater stability by consistently producing similar instance-level feature importance rankings across repeated model fits with different random seeds. Ablation experiments show that each component of LMDI+ contributes to these gains, and that the improvements extend beyond random forests to gradient boosting models. Finally, we show that LMDI+ enables local interpretability use cases by identifying closely matched counterfactuals for each classification benchmark and discovering homogeneous subgroups in a case study using a commonly-used housing dataset.

---

[*]Equal contribution

[1]Implementation is available in the `imodels` package, with experimental code provided in the companion repository.

# 1 Introduction

Despite the growing diversity of data types in various fields, tabular data continues to be one of the most prevalent and widely utilized formats in real-world applications. Tree-based models such as random forests (RFs) and gradient boosting (Freund and Schapire, 1995; Breiman, 2001) have consistently demonstrated state-of-the-art performance and computational efficiency on tabular datasets, often outperforming deep learning-based algorithms (Gorishniy et al., 2021; Shwartz-Ziv and Armon, 2022). Consequently, tree-based models are frequently deployed in high-stakes environments such as healthcare (Haripriya et al., 2021; Loef et al., 2022; Liang et al., 2025) and criminal justice (Tollenaar and Van Der Heijden, 2019; Kovalchuk et al., 2023; Alsubayhin et al., 2024).

However, the use of algorithms to automate decision-making raises concerns about safety, making trustworthy interpretations necessary for deployment in critical settings. Local feature importance (LFI) methods play a central role in this effort by identifying important features that drive each observation's prediction. In practice, LIME (**L**ocal **I**nterpretable **M**odel-agnostic **E**xplanations) and SHAP (**Sh**apley **A**dditive Ex**p**lanations) (Ribeiro et al., 2016; Lundberg and Lee, 2017) are among the most widely used LFI approaches for tree-based models and have been applied in various fields (Shi et al., 2022; Vimbi et al., 2024).

Despite the prevalence of these methods, researchers have demonstrated significant limitations (Alvarez-Melis and Jaakkola, 2018; Zhang et al., 2019; Zafar and Khan, 2019; Bilodeau et al., 2024). In particular, their model-agnostic nature requires measuring changes in predictions of perturbed inputs rather than directly utilizing the inner model structure. Specifically, LIME generates random perturbations around each instance and provides explanations by fitting a sparse linear model over the predicted responses. Similarly, SHAP approximates the change in model output by perturbing feature inclusion across all possible subsets. While such model-agnostic methods are useful when the model internals are inaccessible, they may overlook rich structural information that can produce more faithful explanations. Consequently, these methods have been shown to fail to identify the true signal features in the data-generating processes (DGPs) (Bilodeau et al., 2024; Huang and Marques-Silva, 2024) and produce unstable feature importances (Zafar and Khan, 2019; Zhang et al., 2019).

In contrast to methods that rely on perturbations or feature permutations, tree-based models offer direct access to their internal structure. Mean Decrease in Impurity (MDI) leverages this structure to quantify global feature importance by computing the average reduction in impurity (e.g., Gini or entropy) resulting from splits on each feature (Breiman et al., 1984). Local MDI further extends this idea by tracing impurity reductions along the decision path taken by each instance, thereby attributing importance at the sample level (Sutera et al., 2021). However, tree-based models have been shown to be inefficient at capturing additive signal (Tan et al., 2023), and MDI has well-documented bias where features with high entropy or low correlation with other features tend to receive higher feature importance scores than they should (Strobl et al., 2007; Genuer et al., 2010). Agarwal et al. (2023) thus introduced MDI+, a global feature importance framework for tree-based models which reduce these biases through an enhanced data representation with linear additive structures and regularization. Nevertheless, due to the global nature of MDI+, the resulting feature importance scores cannot be used to explain sample-specific outcomes, which often depend on unique individual characteristics.

To address this gap, we propose Local MDI+ (LMDI+), a novel extension of the MDI+ framework to the sample-specific setting. The development of LMDI+ was driven by the Predictability-Computability-Stability (PCS) framework for veridical data science (Yu and Kumbier, 2020; Yu and Barter, 2024; Rewolinski and Yu, 2025). The PCS framework unifies best practices in data science and machine learning through its namesake core principles. LMDI+ leverages these principles to inherit the strengths of MDI+ while addressing shortcomings of existing LFI methods by more faithfully and consistently identifying signal features in the underlying DGPs. In this work, we demonstrate that LMDI+ accomplishes the following:

1. **Better identifies signal and predictive features.** Across diverse synthetic settings, LMDI+ consistently achieves state-of-the-art performance in identifying instance-specific signal features under varying DGPs, sample sizes, noise levels, and feature correlations. On twelve real-world benchmark datasets, it detects features that are more predictive of individual outcomes.

2. **Ensures stability in feature identification.** Stability experiments show that LMDI+ produces the most consistent feature rankings across repeated model fits with different random seeds, demonstrating superior stability compared to baseline methods.

3. **Generalizes across tree ensembles.** Ablation experiments reveal that each component of LMDI+ contributes to its gains, and that these improvements extend beyond RFs to other tree-based models such as gradient boosting.

4. **Enhances practical applications of LFI.** We show that LMDI+ detects actionable counterfactual explanations which prioritize signal features across six classification benchmarks. Furthermore, we show LMDI+ discovers homogeneous subgroups which simplify the underlying prediction task in a case study on the Miami Housing dataset, leading to better downstream accuracy.

## 2   Related Work

**Tree-based models.** Tree-based models offer a straightforward rule-based approach to prediction and are renowned for their exceptional performance and computational efficiency (Breiman, 2001; Ke et al., 2017; Shwartz-Ziv and Armon, 2022). The most popular decision tree algorithm is CART (classification and regression trees), which generates predictions using a single tree (Breiman et al., 1984). Tan et al. (2025) propose FIGS, an extension of CART which improves upon prediction performance by accounting for additive structure. However, individual trees are weak learners (Quinlan et al., 1996) and thus tend to underperform on complex tasks (Banerjee et al., 2019). Ensemble methods, such as RFs (Breiman, 2001) and variants including generalized RFs (Athey et al., 2019) and iterative RFs (Basu et al., 2018), seek to enhance predictive performance by aggregating multiple trees trained on bootstrapped samples of data.

**Global feature importance.** Global feature importance methods summarize the overall contributions of features across the entire dataset. One approach is permutation-based importance, such as Mean Decrease in Accuracy (MDA) (Breiman, 2001). MDA permutes the values of a column for out-of-bag samples and measures the resulting drop in prediction accuracy. Since these permutations break dependencies between features, MDA performs poorly when features are highly correlated (Hooker and Mentch, 2019; Hooker et al., 2021). Another widely used tree-based global importance method, and more closely related to our work, is Mean Decrease in Impurity (MDI) (Breiman et al., 1984). MDI assigns importance to a feature $k$ according to the decrease in impurity resulting from nodes splitting on $k$. However, MDI is known to favor features with high entropy or low correlation with other features, regardless of their relationship with the outcome (Strobl et al., 2007; Genuer et al., 2010). The MDI+ framework (Agarwal et al., 2023) explains these drawbacks by providing a reinterpretation of MDI as an $R^2$ value in an equivalent linear regression, and further addresses them by using regularized generalized linear models (GLMs) and appending smooth features to enrich the linear representation.

**Local feature importance.** Local feature importances attempt to explain individual predictions, providing values for each observation rather than for the model as a whole. LIME and SHAP are two popular LFI methods which are often used to explain tree-based models. LIME generates random perturbations around each instance and provides explanations by fitting a sparse linear model over the predicted responses (Ribeiro et al., 2016). However, this reliance on random perturbations introduces significant instability due to randomness in the sampling process (Alvarez-Melis and Jaakkola, 2018; Zafar and Khan, 2019; Zhang et al., 2019). SHAP, inspired by game-theoretic Shapley values, approximates the average contribution of a feature across all possible feature permutations (Lundberg and Lee, 2017). TreeSHAP further improves the computational efficiency of this process for tree-based models (Lundberg et al., 2020). However, SHAP and TreeSHAP have been found to fail to identify signal features and can yield misleading feature importance rankings (Huang and Marques-Silva, 2024; Bilodeau et al., 2024). Instead of relying on perturbations or feature permutations, Sutera et al. (2021) propose Local MDI, which extends MDI to local explanations by tracing impurity reductions along the decision path taken by each instance. Plumb et al. (2018) propose MAPLE, which uses random forest leaf assignments to weight training instances when fitting local linear explanation models.

# 3 The Local MDI+ Framework

This section introduces the LMDI+ framework for local feature importance. We begin with a brief review of MDI and subsequently review the connection between MDI and the $R^2$ metric from linear regression, giving rise to MDI+. We then exploit this linear regression interpretation of MDI by extending the MDI+ framework for global feature importance to the sample-specific setting.

## 3.1 Mean Decrease in Impurity

Let dataset $\mathcal{D} = \{(\mathbf{x}_i, y_i)\}_{i=1}^n$ be given, with covariates $\mathbf{x}_i \in \mathbb{R}^p$ and responses $y_i \in \mathbb{R}$. We fit a decision tree by partitioning training data using axis-aligned splits which maximize the decrease in impurity. Specifically, a split $s$ at node $\boldsymbol{v}$ partitions the data into two children nodes $\boldsymbol{v}_L = \{\mathbf{x}_i \in \boldsymbol{v} : x_{i,k} \leq \tau\}$ and $\boldsymbol{v}_R = \{\mathbf{x}_i \in \boldsymbol{v} : x_{i,k} > \tau\}$ for some covariate index $k$ and threshold $\tau$. We define the *impurity decrease* of $s$ to be

$$\hat{\Delta}(s, \mathcal{D}) := N(\boldsymbol{v})^{-1} \left( \sum_{i:\mathbf{x}_i \in \boldsymbol{v}} (y_i - \overline{y}_{\boldsymbol{v}})^2 - \sum_{i:\mathbf{x}_i \in \boldsymbol{v}_L} (y_i - \overline{y}_{\boldsymbol{v}_L})^2 - \sum_{i:\mathbf{x}_i \in \boldsymbol{v}_R} (y_i - \overline{y}_{\boldsymbol{v}_R})^2 \right), \tag{1}$$

where $N(\boldsymbol{v})$ denotes the number of training samples in node $\boldsymbol{v}$ and $\overline{y}_{\boldsymbol{v}}$ denotes the mean response of training samples in node $\boldsymbol{v}$. For a tree with *structure* $\mathcal{S} = \{s_1, \ldots, s_m\}$, the MDI of feature $X_k$ is defined as

$$MDI_k(\mathcal{S}, \mathcal{D}) := \sum_{s \in \mathcal{S}^{(k)}} n^{-1} N(\boldsymbol{v}(s)) \hat{\Delta}(s, \mathcal{D}), \tag{2}$$

where $\mathcal{S}^{(k)}$ represents the splits which threshold feature $X_k$ and $\boldsymbol{v}(s)$ is the node split by $s$.

## 3.2 Connecting MDI to $R^2$ Values from Linear Regression

Using the notation described above, we define the stump function

$$\psi(\mathbf{x}_i; s) = \frac{N(\boldsymbol{v}_R) \mathbf{1}\{x_i \in \boldsymbol{v}_L\} - N(\boldsymbol{v}_L) \mathbf{1}\{x_i \in \boldsymbol{v}_R\}}{\sqrt{N(\boldsymbol{v}_L) N(\boldsymbol{v}_R)}}. \tag{3}$$

Note that $\psi$ takes three values, corresponding to whether the observation $x_i$ is in the left child, the right child, or not in node $\boldsymbol{v}$. Concatenating the stump functions for all $m$ splits in the tree on $\mathbf{X} \in \mathbb{R}^{n \times p}$ results in feature map $\Psi(\mathbf{X}; \mathcal{S}) := (\psi(\mathbf{X}; s_1), \ldots, \psi(\mathbf{X}; s_m)) \in \mathbb{R}^{n \times m}$. Klusowski and Tian (2023) showed that the fitted decision tree model predictions are equivalent to those of the ordinary least squares (OLS) model by regressing the responses $\mathbf{y}$ on $\Psi(\mathbf{X}; \mathcal{S})$. Agarwal et al. (2023) further build on this connection between decision trees and linear models by deriving a connection between MDI and $R^2$, stated next.

**Proposition 1 (Agarwal et al. (2023))** *Assume a tree fit on bootstrapped dataset $\mathcal{D}^* = (\mathbf{X}^*, \mathbf{y}^*)$ has structure $\mathcal{S}$. Then, for feature $X_k$, we have:*

$$MDI_k(\mathcal{S}, \mathcal{D}^*) \propto R^2(\mathbf{y}, \hat{\mathbf{y}}^{(k)}), \tag{4}$$

*where $\hat{\mathbf{y}}^{(k)} \in \mathbb{R}^n$ is the vector of predicted responses from the OLS regression $\mathbf{y}^* \sim \Psi(\mathbf{X}^*; \mathcal{S}^{(k)})$.*

This relationship helps to illuminate several drawbacks of MDI:

1. **Overfitting.** Proposition 1 shows that MDI is implicitly evaluated as a prediction $R^2$ using the same in-bag samples that were used to train the decision tree model. Evaluating this predictive metric on the training data constitutes "data snooping", which inevitably leads to overfitting and, consequently, produces unreliable feature importance estimates.

2. **Bias against correlated and low-entropy features.** CART favors high-entropy features because they are more likely to produce large impurity reductions by chance (Strobl et al., 2007), leading to inflated importance scores. When multiple features share predictive information, CART typically selects only one for an early split, inflating its importance while deflating the scores of remaining correlated features.

3. **Bias against smooth and additive functions.** Proposition 1 shows that the prediction $R^2$ is computed using predictions obtained by regressing on stump features $\Psi(\mathbf{X})$, where each split maps a feature into only three discrete values. This coarse, piecewise-constant representation is therefore inefficient to capture smooth and additive functions (Tsybakov, 2008; Tan et al., 2021), even though such functions are common in real-world data (Hastie and Tibshirani, 1986).

The MDI+ framework (Agarwal et al., 2023) for global feature importance mitigates these drawbacks through the use of out-of-bag (OOB) samples and regularized GLMs, but falls short of being able to explain individual predictions. See Appendix A for a detailed overview of MDI+.

### 3.3 The Local MDI+ Framework

Note that in a linear model with coefficients $\hat{\boldsymbol{\beta}}$, we can discern the exact amount that the $k$th feature contributed to the observation $\mathbf{x}$'s prediction by taking $\hat{\beta}_k x_k$. Recall from Section 3.2 that the predictions of a decision tree are equivalent to those of an OLS model defined over a transformed basis. This equivalence allows us to interpret a tree model through its linearized form and attribute the contribution of individual features in the same way as in a linear model.

We leverage this fact to extend MDI+ to the sample-specific setting, where we propose LMDI+ as follows.

1. **Obtain expanded representation.** Each tree in an RF is fit on a bootstrapped dataset $\mathcal{D}^* = (\mathbf{X}^*, \mathbf{y}^*)$, and MDI is only calculated on the in-bag samples $\Psi(\mathbf{X}^*; \mathcal{S})$. LMDI+ instead appends the raw feature $\mathbf{x}_k \in \mathbb{R}^n$ to the feature map consisting of both in-bag and OOB samples, yielding the transformed representation $\tilde{\Psi}^{(k)}(\mathbf{X}) = \tilde{\Psi}\left(\mathbf{X}; \mathcal{S}^{(k)}\right) = [\Psi\left(\mathbf{X}; \mathcal{S}^{(k)}\right), \mathbf{x}_k]$.

2. **Fit regularized GLM.** Instead of using OLS, fit a regularized GLM $\mathcal{G}$ with link function $g$ and penalty $\lambda$ by regressing response $\mathbf{y}$ on the transformed data $\tilde{\Psi}(\mathbf{X}) = \tilde{\Psi}(\mathbf{X}; \mathcal{S})$.

3. **Calculate feature attributions.** Let $\hat{\boldsymbol{\beta}}_\lambda^{(k)}$ represent the GLM coefficients corresponding to the features in $\tilde{\Psi}^{(k)}(\mathbf{X})$. For each observation $\mathbf{x}$, we then define the LMDI+ of the $k$th feature to be $LMDI_k^+\left(\mathbf{x}, \mathcal{S}^{(k)}, \mathcal{G}\right) := \tilde{\Psi}^{(k)}(\mathbf{x})^\top \hat{\boldsymbol{\beta}}_\lambda^{(k)}$ if $\mathcal{S}^{(k)} \neq \emptyset$, and zero otherwise, since a feature that never appears in any split contributes no structural information to that tree's predictions.

After computing $LMDI_k^+(\mathbf{x}, \mathcal{S}^{(k)}, \mathcal{G})$ for $k = 1, \ldots, p$, we rank features by the absolute value of their importance. The LMDI+ scores of an ensemble are obtained by averaging the scores of its trees.

Local MDI+ leverages the linear representation of the decision tree, where the inner product between the estimated GLM coefficients and the transformed feature values at a given sample yields a direct attribution of feature contributions. By augmenting OLS with expanded feature representation and generalized linear models, LMDI+ inherits the key benefits of the MDI+ framework (Agarwal et al., 2023), including reduced overfitting, correlation bias, and bias against smooth or additive models, addressing the weaknesses of LFI methods that rely on MDI, such as Local MDI (Sutera et al., 2021). Moreover, the linearity of Local MDI+ results in stable and interpretable attributions, eliminating the need for local linear approximations as used in methods like LIME (Ribeiro et al., 2016).

**Outline of results.** The following sections present experiments evaluating the performance of LMDI+. Section 4 demonstrates LMDI+ effectively captures signal features across a wide range of settings. Section 5 shows that LMDI+ produces stable feature importance rankings across different model fits. Section 6 conducts an ablation study to assess the contribution of each component in the LMDI+ framework and evaluates the robustness of LMDI+ by ablating different ensemble settings. Section 7 and Section 8 describe how LMDI+ enhances two practical use cases for local feature importance. Section 9 provides a runtime analysis of LMDI+ relative to competing methods.

# 4  LMDI+ Better Captures Signal Features

Trustworthy interpretations should reflect the predictivity of features in the underlying DGPs (Murdoch et al., 2019; Yu and Kumbier, 2020). In this section, we demonstrate that LMDI+ better differentiates between signal and non-signal features at the sample level across a variety of datasets and DGPs. We first consider the simulation settings (Sections 4.1 and 4.2), where signal features are defined by the ground-truth DGP. We then turn to real-world benchmark datasets (Section 4.3), where we assess how well each method identifies predictive features by measuring performance when retaining only its top-ranked features, varying the proportion kept from 10% to the full set. We next introduce the datasets and baselines.

**Datasets.** We selected commonly-used benchmarks datasets from the OpenML repository (Vanschoren et al., 2013). For regression tasks, we use datasets from the CTR23 curated tabular regression benchmark (Fischer et al., 2023). For classification tasks, we use datasets from Grinsztajn et al. (2022)'s benchmark and the CC18 curated classification benchmark (Bischl et al., 2019). We selected six regression datasets and six classification datasets with a large number of features from the repository, and consistently using these datasets across all experiments. Details of the selected datasets including preprocessing, train-test split, and feature counts, are provided in Appendix B.

**Implementation Details.** All models are implemented in `scikit-learn` (Pedregosa et al., 2011). For regression tasks, we use `RandomForestRegressor` with `n_estimators=100`, `min_samples_leaf=5`, and `max_features=0.33`. For classification tasks, we use `RandomForestClassifier` with `n_estimators=100`, `min_samples_leaf=1`, and `max_features="sqrt"`. We additionally vary these hyperparameters in the ablation study (Section 6) to demonstrate robustness. When implementing LMDI+, we pick ElasticNet as the GLM due to its combination of $\ell_1$ and $\ell_2$ penalties. For regression tasks, we use `ElasticNetCV` with `cv=3` and `l1_ratio=[0.1,0.5,0.99]`. For classification tasks, we use `LogisticRegressionCV` with `penalty="elasticnet"`, `l1_ratios=[0.1,0.5,0.99]` and `cv=3`.

**Baselines.** We compare LMDI+ with commonly used LFI methods, including LIME (Ribeiro et al., 2016) and TreeSHAP (Lundberg et al., 2020), and additionally compare against Local MDI (Sutera et al., 2021) to highlight the improvements of our approach.

## 4.1  Data-Inspired Synthetic Experiments

Evaluating LFI methods is challenging due to the absence of ground-truth feature importances in real-world datasets. To overcome this, we begin by assessing each method's ability to distinguish signal from non-signal features in synthetic settings where the true importance is known. To make these simulations more realistic, we use real-world covariates to capture naturally occurring correlation patterns and noise and generate responses via analytic functions that encompass both linear and nonlinear structures.

**Setup.** In each simulation, a subset of features is randomly selected from each dataset described above to serve as signal features, which are then incorporated into the following response function:

1. Linear model: $\mathbb{E}[Y \mid X] = \sum_{m=1}^{5} X_m$

2. Polynomial interaction model: $\mathbb{E}[Y \mid X] = \sum_{m=1}^{5} X_{2m-1} + \sum_{m=1}^{5} X_{2m-1}X_{2m}$

3. Linear + locally-spiky-sparse (LSS): $\mathbb{E}[Y|X] = \sum_{m=1}^{5} X_{2m-1} + \mathbb{1}(X_{2m-1} > 0)\mathbb{1}(X_{2m} > 0)$

Classification simulations then pass $\mathbb{E}[Y|X]$ through a logistic link function to generate binary responses. These response functions were selected to reflect real-world DGPs. Linear and interaction models are widely studied models in the machine learning literature (Tsang et al., 2021; Tai, 2021), while the Linear+LSS model is commonly observed in fields like genomics (Basu et al., 2018; Behr et al., 2022).

**Parameters.** We vary both the signal-to-noise ratio (SNR) and the sample size. For regression, we measure the SNR by proportion of variance explained ($PVE = \frac{\text{Var}(\mathbb{E}[Y|X])}{\text{Var}(Y)}$). We then take $PVE \in \{0.1, 0.2, 0.4, 0.8\}$.

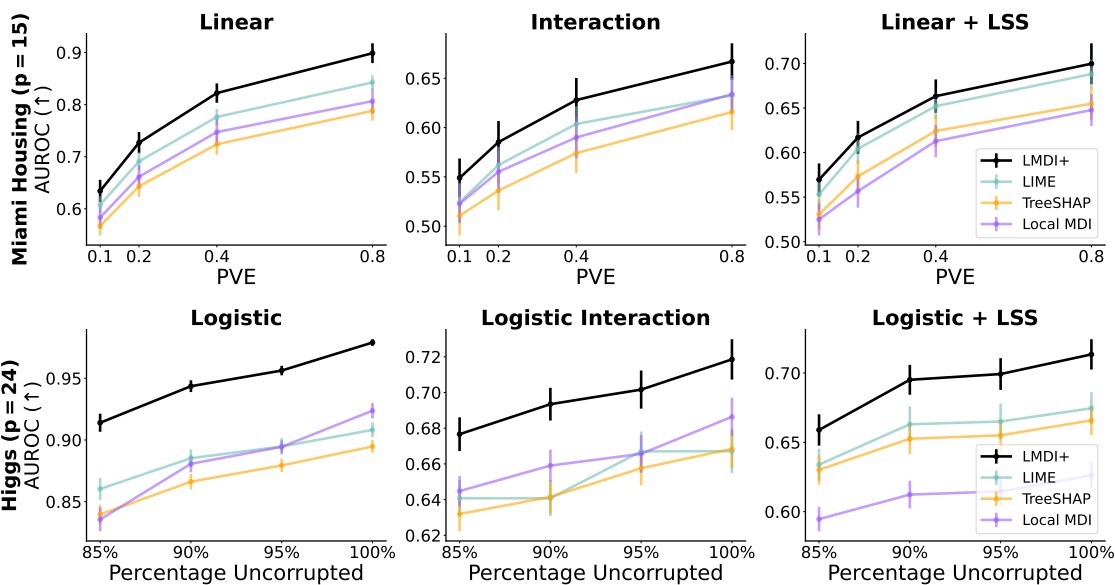

Figure 1: LMDI+ consistently achieves higher AUROC ($\pm 1$ SE) across different datasets, response functions and SNRs, demonstrating its superior ability to distinguish signal features from non-signal features. Results are averaged over 30 runs.

For classification, noise levels are adjusted by randomly flipping $\{0\%, 5\%, 10\%, 15\%\}$ of the labels. Sample sizes are varied over $\{300, 500, 1000\}$.

**Evaluation.** We evaluate the task based on how well each LFI method classifies signal features over non-signal features. We treat this as a classification problem, where signal features are labeled as 1 (positive class) and noise features as 0 (negative class). We compute the AUROC score between the LFI scores and the signal labels for each test sample. Higher AUROC across instances indicates better performance.

**Results.** Figure 1 presents the results on two benchmark datasets with sample size 300. LMDI+ consistently achieves higher AUROC across different response functions and SNRs. LMDI+'s superior ability to distinguish signal from noise is consistent across different datasets and sample sizes. Complete results for all datasets and sample sizes are provided in Appendix C.

## 4.2 LMDI+ is Robust to Strong Correlation

Existing LFI methods struggle to identify signal features when they are highly correlated with noisy ones. In this section, we show that LMDI+ is robust to strong correlation.

**Setup.** We follow the setup detailed in Section 7.1 of Agarwal et al. (2023). Specifically, we draw 250 samples $\mathbf{x} \sim \mathcal{N}_{100}(\mathbf{0}, \mathbf{\Sigma})$, where $\mathbf{\Sigma}$ has a block structure: features $X_1, \ldots, X_{50}$ share pairwise correlation $\rho$, while $X_{51}, \ldots, X_{100}$ are independent of all other features. Responses $y$ are generated using a Linear+LSS model (see Section 4.1), with signal features $X_1, \ldots, X_6$.

**Parameters.** We fix $PVE = 0.1$ and vary correlation $\rho \in \{0.5, 0.6, 0.7, 0.8, 0.9, 0.99\}$.

**Evaluation.** We divide the features into three groups: (1) signal features, (2) non-signal features that are correlated with signal, and (3) independent non-signal features. For each LFI method, we compute the average feature ranking of each group. Robust methods should consistently rank signal features as more important than both types of non-signal features, regardless of correlation strength.

**Results.** Figure 2 displays the average per sample rank of each feature group according to various LFI methods. We observe that LIME, TreeSHAP, and Local MDI incorrectly rank uncorrelated non-signal features as more

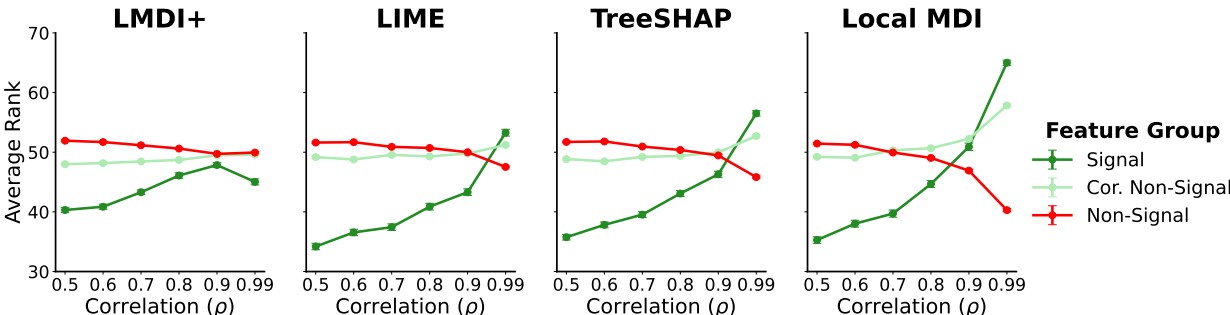

Figure 2: We show the average per sample ranks ($\pm 1$ SE) of each feature group under various levels of correlation $\rho$. In the presence of extreme correlation, only LMDI+ continues to rank the signal features as most important. Results are averaged over 50 runs.

important than signal features in the presence of strong correlation. However, the use of regularized GLMs and OOB samples helps LMDI+ rank signal features as the most important across all levels of correlation.

### 4.3 Real Data Experiments

We now turn to evaluation using the real datasets described in Table 7. Recent work has highlighted the challenges of evaluating feature attributions (Monteiro Paes et al., 2024; Edin et al., 2025); since ground truth is unavailable, we adopt a commonly used remove-and-retrain evaluation technique (Hooker et al., 2019; Alangari et al., 2023) to demonstrate that LMDI+ identifies features that are more predictive of individual outcomes.

**Setup.** For each observation, we kept only the top $k\%$ of features according to individual LFI rankings, and replaced all remaining features with their average. An RF model was then retrained on this masked training set and evaluated on the original test set. Due to the prohibitive runtime of LIME on large datasets, datasets with more than 2,000 samples were downsampled to 2,000 for full comparison. Additional experiments on full datasets excluding LIME are presented in Appendix D , and a detailed runtime comparison is provided in Section 9.

**Parameters.** We vary the proportion $k \in \{10\%, 20\%, \ldots, 100\%\}$ of features retained per sample, by selecting the top $k\%$ features based on individual feature importance rankings.

**Evaluation.** A better LFI method should identify more predictive features and therefore achieve higher performance after retraining. We report $R^2$ for regression and AUROC for classification.

**Results.** Figure 3 shows performance on the *Puma Robot* and *SARCOS* datasets (regression) and the *Higgs* and *Ozone* datasets (classification). LMDI+ achieves higher $R^2$ and AUROC after retraining on the masked data, demonstrating its ability to identify more predictive features. The performance gap between LMDI+ and the baselines becomes even more pronounced when a smaller proportion of features is retained. We summarize the full results across all 12 datasets in Table 1, reporting the average rank of each method. LMDI+ consistently achieves the top rank across different feature selection levels. Results for all datasets are provided in Appendix D.

| | **Average Rank** ($\downarrow$) | | | |
|---|---|---|---|---|
| | Top Features Retained | | | |
| Method | 10% | 20% | 30% | 40% |
| LMDI+ | **1.25** | **1.17** | **1.00** | **1.08** |
| LIME | 2.17 | 2.17 | 2.42 | 2.33 |
| TreeSHAP | 2.83 | 2.83 | 2.83 | 3.08 |
| Local MDI | 3.75 | 3.83 | 3.75 | 3.50 |

Table 1: We report the average rank of methods across 12 datasets, where lower ranks (closer to 1) indicate better performance. LMDI+ consistently achieves the best rank across all feature proportions, identifying more predictive features. Results are averaged over 20 runs per dataset.

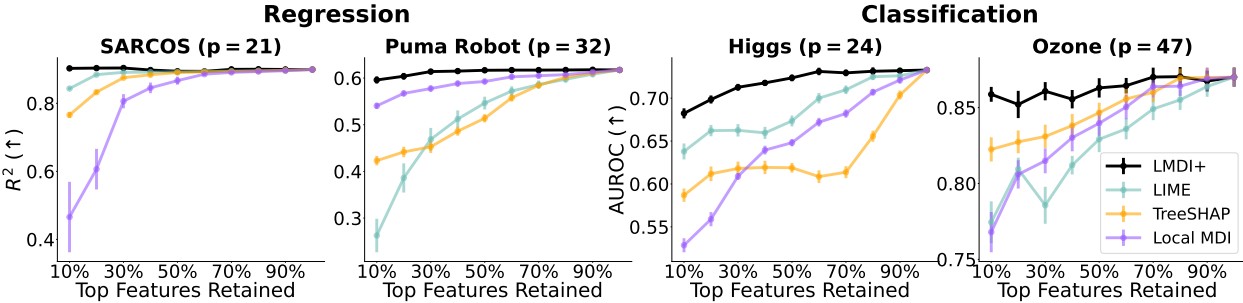

Figure 3: LMDI+ achieves higher $R^2$ and AUROC after retraining on masked training data, demonstrating its superior ability to identify predictive features. Results are averaged over 20 runs.

# 5 LMDI+ Produces More Stable Feature Importance Rankings

Trustworthy interpretations require robustness to choices made during the modeling process (Murdoch et al., 2019; Yu and Kumbier, 2020; Burger et al., 2023). A well-known limitation of LFI methods is their instability (Zafar and Khan, 2019; Zhang et al., 2019). Different runs of the model yield different feature rankings, leading to inconsistent and unreliable explanations. In this section, we show that LMDI+ consistently produces more stable feature rankings across repeated model fits with different random seeds, addressing the instability of existing methods.

**Setup.** For each dataset described in Table 7, we trained five RF models, each initialized with a different random seed. For each sample, we identified the top $k\%$ most important features according to each LFI method and counted the number of unique features selected across the five fitted RF models.

**Parameters.** We vary the proportion of features retained per sample by selecting the top $k \in \{10\%, 20\%, 30\%, 40\%\}$ based on individual feature importance rankings. In addition, we vary the sample size by subsampling $\{500, 1000, 2000\}$ instances from each dataset.

**Evaluation.** We measure the number of unique features selected for each sample across the five RF fits, normalized by the total number of features. Lower values indicate that the same features are consistently selected across model fits, reflecting more stable and robust feature rankings.

**Results.** We present the results for *Puma Robot* and *SARCOS* datasets (regression) and the *Higgs* and *Ozone* datasets (classification) in Figure 4. Across varying sample sizes and percentages of selected features, LMDI+ consistently identifies the smallest number of unique features, indicating greater stability compared to the baselines. The complete results across all datasets and sample sizes are provided in Appendix E, where LMDI+ consistently achieves the greatest stability.

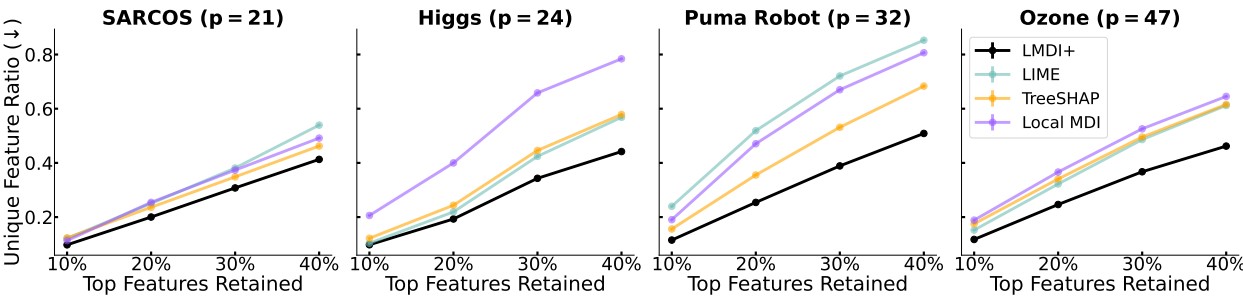

Figure 4: LMDI+ selects the fewest unique features, indicating the most consistent feature rankings across repeated RF fits. Results are averaged over 15 runs.

## 6 Ablation Analysis

As described in Section 3.3, our method builds on three key components: the use of out-of-bag (OOB) samples, feature transformations, and generalized linear models (GLMs). In this section, we start by conducting an ablation analysis to evaluate the contribution of each of these components. We report results from real data feature selection experiments (Section 4.3) and stability experiments (Section 5), illustrating how each component helps address the known limitations of the baseline methods.

Table 2 summarizes the performance as each component is added. We report the average rank across all datasets, where a lower rank (closer to 1) indicates better performance. The ablation study reveals a consistent performance gain with each added component, and the full LMDI+ achieves the highest overall ranking, demonstrating that all components contribute meaningfully to performance.

Beyond the default RF configuration, we further assess the robustness of LMDI+ by ablating different ensemble settings. Specifically, we vary the number of estimators (`n_estimators` $\in \{10, 50\}$) and the minimum number of samples per leaf (`min_samples_leaf` $\in \{5, 10\}$). In addition, we evaluate LMDI+ with gradient boosting, another widely used tree-based model in practice, to demonstrate its applicability beyond RFs.

Table 3 presents the results for RF with `min_samples_leaf` = 5 and gradient boosting configurations, while additional results for other settings are presented in Appendix F. Across all configurations, LMDI+ achieves the best average performance in feature selection and stability across all twelve datasets, demonstrating that LMDI+ is robust to changes in ensemble structure. Importantly, its strong performance with gradient boosting models shows that LMDI+ extends naturally beyond RFs to widely used boosting-based algorithms.

| | | | | Average Rank ($\downarrow$) | | | |
|---|---|---|---|---|---|---|---|
| | | | | | Percent of Features Retained | | |
| | | Components | | | Feature Selection | | Stability |
| LMDI+ Variant | OOB | Raw Features | GLMs | 10% | 20% | 10% | 20% |
| 1 | ✗ | ✗ | ✗ | 2.50 | 2.00 | 3.08 | 3.00 |
| 2 | ✓ | ✗ | ✗ | 1.67 | 1.83 | 2.75 | 2.42 |
| 3 | ✓ | ✓ | ✗ | 1.67 | **1.17** | 1.33 | 1.17 |
| LMDI+ | ✓ | ✓ | ✓ | **1.25** | **1.17** | **1.17** | **1.00** |

Table 2: Ablation study for LMDI+ components. Results show the average rank of each ablated LMDI+ variant, computed over TreeSHAP, LIME, and Local MDI on all twelve datasets, where lower ranks (closer to 1) indicate better performance.

| | | Average Rank ($\downarrow$) | | | |
|---|---|---|---|---|---|
| | | Percent of Features Retained | | | |
| | | Feature Selection | | Stability | |
| Ensemble | Method | 10% | 20% | 10% | 20% |
| Random Forest (`min_sample_leaf` = 5) | LMDI+ | **1.25** | **1.08** | **1.17** | **1.00** |
| | LIME | 2.50 | 2.33 | 2.92 | 2.83 |
| | TreeSHAP | 2.58 | 2.92 | 2.42 | 2.42 |
| | Local MDI | 3.67 | 3.67 | 3.58 | 3.25 |
| Gradient Boosting | LMDI+ | **1.08** | **1.08** | **1.17** | **1.00** |
| | LIME | 2.33 | 2.25 | 2.75 | 2.75 |
| | TreeSHAP | 2.58 | 2.67 | 2.25 | 2.25 |

Table 3: Ablation study for LMDI+ across different ensembles. Results show the average rank of methods across all twelve datasets. Local MDI is not applicable to gradient boosting and is therefore omitted.

## 7 LMDI+ Finds Practical Counterfactuals

Counterfactual explanations describe how an observation would need to change to obtain a different classification. They are typically computed on the observed data and thus treat all features equally, regardless of importance (Molnar, 2025). In Appendix G, we show that calculating counterfactuals on the LFI matrix yields explanations that prioritize proximity with respect to signal features. In this section, we demonstrate that LMDI+ produces improved counterfactual explanations by preferring features with high local importance.

**Setup.** For each classification dataset in Table 7, we train an RF and compute LFI on both the training and test data. Counterfactuals are traditionally computed by finding the minimum-norm vector that crosses the decision boundary (Wachter et al., 2017; Dandl et al., 2020). However, this can lead to impossible feature values. To avoid such issues, we instead match the observation to the most similar training example using a kNN approach (Fix, 1985; Yadav and Chaudhuri, 2021; Brughmans et al., 2024). Specifically, for each test sample $\mathbf{x}$ with predicted label $\hat{y}$ and feature importance vector $\mathbf{w}$, we find the training sample $\mathbf{x}_j$ which minimizes $||\mathbf{w} - \mathbf{w}_j||_1$ subject to $\hat{y} \neq \hat{y}_j$.

**Evaluation.** Good counterfactuals aim to be as similar as possible to the original observation, thus minimizing unnecessary changes. To measure similarity, we use the $\ell_1$ distance (as suggested by Wachter et al. (2017)) $||\mathbf{x} - \mathbf{x}_j||_1$ between the counterfactual and the test sample.

**Results.** Table 4 shows the mean $\ell_1$ distance to the counterfactuals detected by each LFI method. We observe that for every dataset, LMDI+ has the smallest average distance, thus minimizing the amount of change necessary to flip the predictions. This means that LMDI+ identifies counterfactuals that require smaller modifications to reach the desired outcome.

| Mean $\ell_1$ Distance to Counterfactual ($\downarrow$) | | | | | | |
|---|---|---|---|---|---|---|
| | House 16H | Higgs | Pol | Ozone | Jannis | Spam |
| LMDI+ | **7.6** | **18.2** | **9.1** | **34.7** | **40.4** | **15.2** |
| LIME | 9.5 | 19.9 | 11.4 | 36.3 | 44.7 | 18.7 |
| TreeSHAP | 9.5 | 20.6 | 10.9 | 44.3 | 44.3 | 19.8 |
| Local MDI | 12.7 | 22.8 | 11.2 | 39.0 | 48.1 | 21.1 |

Table 4: LMDI+ detects closer counterfactual explanations than LIME, TreeSHAP, and Local MDI.

## 8 Case Study: LMDI+ Discovers More Homogeneous Subgroups

Subgroup identification allows practitioners to better understand patterns which exist in their data. Particularly, clustering algorithms partition observations into subgroups which may share common traits. In this section, we investigate how using LFI scores to form homogeneous subgroups can simplify the underlying prediction task.

**Setup.** We select *Miami Housing* from the benchmarks listed in Table 7 due to its easily understandable task (predict sale price) and features (location, size, etc.). Using a 50/50 train-test split, we fit an RF on the training data and calculate LFI scores on the test data. We then partition the data into subgroups by performing $k$-means clustering (for $k = 2, \ldots, 10$) on each method's LFI matrix.

**Evaluation.** We fit an OLS regression on each subgroup detected by an LFI method. If the LFI method has discovered more homogeneous subgroups, the simple OLS models that were fitted per subgroup should yield better prediction performance than the global OLS model fitted on the full test. We examine the mean squared error of the regression models across $k = 2, \ldots, 10$ for each method.

**Results.** Figure 5 displays how subgroup selection changes the error of sale price prediction. The constant "Global" line represents the error of OLS fit to the entire dataset. We observe that clustering on LFI scores achieves lower error than fitting a model on the full data, with LMDI+ performing 40% better than the 'Global' model. Among LFI methods, LMDI+ consistently yields the lowest error, reaching an elbow point at $k = 4$ while Local MDI takes longer to achieve low MSE. Furthermore, the MSE from the clusters generated by LMDI+ continues to decrease as $k$ increases, whereas TreeSHAP's clusters see a slight increase for large $k$. Appendix H details an exploration of the subgroups formed by clustering on LMDI+ scores, highlighting geographic compactness and differentiation of important features.

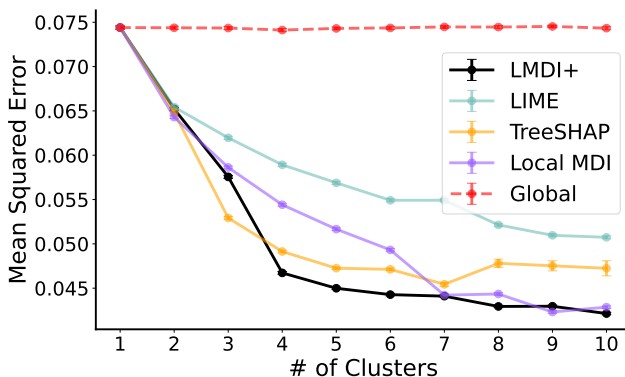

Figure 5: We show the MSE ($\pm 1 SE$) of linear regressions fit on each cluster. We observe a steep drop until $k = 4$, after which MSE levels off.

## 9 Runtime Analysis

In this section, we compare the runtime of LMDI+ to that of other LFI methods. The computation time for obtaining LMDI+ scores can be decomposed into two parts:

1. Obtaining the transformed representation described in Step 1 of Section 3.3 and fitting the GLM (a one-time overhead cost). The transformed dataset contains one feature for each internal node in the tree, so the runtime increases with tree depth.

2. After obtaining the GLM coefficients, LMDI+ scores for test samples are computed quickly via simple linear multiplication with the fitted coefficients.

We also note that LMDI+ is parallelizable across trees, allowing it to gracefully handle large ensembles given sufficient compute. Runtime results for differing tree depths and ensemble sizes can be seen in Table 5 and Table 6, respectively. The runtime experiments were conducted using 16 CPU cores. As described in Section 4.3, we downsample each dataset to 2,000 samples due to the intractability of obtaining LIME scores on the full datasets. Unless specified otherwise in the tables, we use the parameters from Section 4.

We observe that Local MDI is consistently the most computationally efficient method, as its implementation computes local importances concurrently with predictions, incurring no additional cost. For trees that are

| Dataset | # of Features | Min. Samples per Leaf | LMDI+ | LIME | TreeSHAP | Local MDI |
|---|---|---|---|---|---|---|
| SARCOS | 21 | 1 | 0.0745 | 0.1323 | 0.0627 | 0.0016 |
| Puma Robot | 32 | 1 | 0.1241 | 0.1733 | 0.0851 | 0.0016 |
| Wave Energy | 48 | 1 | 0.1036 | 0.2049 | 0.0947 | 0.0015 |
| Super Conductivity | 81 | 1 | 0.1501 | 0.2569 | 0.0907 | 0.0014 |
| SARCOS | 21 | 5 | 0.0113 | 0.1050 | 0.0068 | 0.0011 |
| Puma Robot | 32 | 5 | 0.0175 | 0.1423 | 0.0089 | 0.0013 |
| Wave Energy | 48 | 5 | 0.0227 | 0.1789 | 0.0095 | 0.0011 |
| Super Conductivity | 81 | 5 | 0.0448 | 0.2351 | 0.0093 | 0.0011 |
| SARCOS | 21 | 10 | 0.0081 | 0.0993 | 0.0028 | 0.0010 |
| Puma Robot | 32 | 10 | 0.0128 | 0.1332 | 0.0031 | 0.0010 |
| Wave Energy | 48 | 10 | 0.0165 | 0.1716 | 0.0038 | 0.0009 |
| Super Conductivity | 81 | 10 | 0.0318 | 0.2275 | 0.0033 | 0.0009 |

Table 5: Per-sample runtime (seconds) while varying maximum tree depth.

| Dataset | # of Features | # of Estimators | LMDI+ | LIME | TreeSHAP | Local MDI |
|---|---|---|---|---|---|---|
| SARCOS | 21 | 100 | 0.0113 | 0.1050 | 0.0068 | 0.0011 |
| Puma Robot | 32 | 100 | 0.0175 | 0.1423 | 0.0089 | 0.0013 |
| Wave Energy | 48 | 100 | 0.0227 | 0.1789 | 0.0095 | 0.0011 |
| Super Conductivity | 81 | 100 | 0.0448 | 0.2351 | 0.0093 | 0.0011 |
| SARCOS | 21 | 500 | 0.0600 | 0.2585 | 0.0345 | 0.0054 |
| Puma Robot | 32 | 500 | 0.0859 | 0.3133 | 0.0400 | 0.0058 |
| Wave Energy | 48 | 500 | 0.1171 | 0.3416 | 0.0486 | 0.0054 |
| Super Conductivity | 81 | 500 | 0.2256 | 0.3800 | 0.0447 | 0.0051 |
| SARCOS | 21 | 1000 | 0.1721 | 0.4315 | 0.0677 | 0.0107 |
| Puma Robot | 32 | 1000 | 0.2230 | 0.5056 | 0.0800 | 0.0114 |
| Wave Energy | 48 | 1000 | 0.3602 | 0.5364 | 0.1018 | 0.0112 |
| Super Conductivity | 81 | 1000 | 0.4961 | 0.5497 | 0.0926 | 0.0106 |

Table 6: Per-sample runtime (seconds) while varying ensemble size.

not grown to full depth, LMDI+ is faster than LIME but slower than TreeSHAP. When trees are fully grown, LIME is noticeably slower than LMDI+, while TreeSHAP performs comparably. Finally, the computation time of LMDI+ increases as the ensemble grows, with a similar trend observed for LIME and TreeSHAP. Across ensemble sizes, LMDI+ is consistently faster than LIME but slower than TreeSHAP.

## 10 Discussion

LMDI+ is a sample-specific extension of MDI+, leveraging the connection between decision trees and least squares to combine linear and tree-based notions of feature importance without relying on perturbations or approximations. LMDI+ was developed in adherence to the PCS framework for veridical data science (Yu and Kumbier, 2020; Yu and Barter, 2024), creating an emphasis on generating LFIs that consistently identify instance-specific signal features in the underlying DGPs. Across a wide range of experimental setups and datasets, LMDI+ outperforms other LFI methods in identifying truly important features and produces more stable feature rankings under repeated model fits with different random seeds. We show that each component of LMDI+ contributes to improved identification of signal features, and that this strength of identification can be used to improve counterfactual explanations and subgroup discovery. This makes LMDI+ particularly valuable in high-stakes applications where trustworthy local explanations are critical.

**Limitations.** As mentioned in Section 9, whether or not the computational complexity of LMDI+ is a limitation depends on the applied use case. If the intake of new test points requires the user to update their model, thus necessitating a recalculation of the node basis and refitting of the GLM, then efficiently producing LMDI+ scores will be challenging. However, if the practitioner is able to keep their GLM fixed as new test points arrive, LMDI+ scores reduce to a simple dot product, making inference highly efficient. Another potential limitation of our empirical results is the breadth of the ablation studies in Section 6. While these studies demonstrate robustness across several RF configurations and gradient boosting, a limited subset of possible ensemble settings are covered, and a more exhaustive evaluation of robustness across ensemble structures remains an important direction for future work.

**Future Work.** LMDI+ offers many opportunities for future extensions. While we demonstrate it on RFs and gradient boosting, LMDI+ can be extended to many additional tree-based algorithms. Its flexible framework also supports 0choices of generalized linear models and transformations for continuous features. Future work could also include characterizing the impact of interactions as well as exploring the use of deep learning models instead of GLMs.

## Acknowledgments and Disclosure of Funding

Partial support is gratefully acknowledged from NSF grant DMS-2515767, NSF grant DMS-2413265, NSF grant DMS 2209975, NSF grant 2023505 on Collaborative Research: Foundations of Data Science Institute (FODSI), the NSF and the Simons Foundation for the Collaboration on the Theoretical Foundations of Deep Learning through awards DMS-2031883 and 814639, NSF grant MC2378 to the Institute for Artificial CyberThreat Intelligence and OperatioN (ACTION), and NIH grant R01GM152718. ZTR is supported by the National Science Foundation Graduate Research Fellowship Program under Grant No. DGE-2146752. Any opinions, findings, and conclusions or recommendations expressed in this material are those of the authors and do not necessarily reflect the views of the National Science Foundation.

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

## A  The MDI+ Framework

MDI+ allows for a more flexible modeling framework, reducing the biases of MDI through additions such as out-of-bag samples and regularized GLMs (Agarwal et al., 2023). Particularly, MDI+ does the following:

1. **Obtain enhanced representation.** Each tree in an RF is fit on a bootstrapped dataset $\mathcal{D}^* = (\mathbf{X}^*, \mathbf{y}^*)$. MDI regresses only on in-bag samples $\Psi(\mathbf{X}^*; \mathcal{S})$. MDI+ instead appends the raw feature $\mathbf{x}_k \in \mathbb{R}^n$ to the feature map consisting of both in-bag and out-of-bag samples, yielding the transformed representation $\tilde{\Psi}^{(k)}(\mathbf{X}) = \tilde{\Psi}(\mathbf{X}; \mathcal{S}^{(k)}) = [\Psi(\mathbf{X}; \mathcal{S}^{(k)}), \mathbf{x}_k]$.

2. **Fit regularized GLM.** Instead of using OLS, fit a regularized GLM $\mathcal{G}$ with link function $g$ and penalty $\lambda$ by regressing response $\mathbf{y}$ on the transformed data $\tilde{\Psi}(\mathbf{X}) = \tilde{\Psi}(\mathbf{X}; \mathcal{S})$.

3. **Make partial model predictions.** Let $\bar{\tilde{\Psi}}^{(j)}$ denote the vector of average values for features in $\tilde{\Psi}^{(j)}(\mathbf{X})$. For $k = 1, \ldots, p$, we then define the partial model predictions for each sample $\mathbf{x}_i$ to be

$$\hat{y}_i^{(k)} = g^{-1}\left(\left[\bar{\tilde{\Psi}}^{(1)}, \ldots, \bar{\tilde{\Psi}}^{(k-1)}, \tilde{\Psi}^{(k)}(\mathbf{x}_i), \bar{\tilde{\Psi}}^{(k+1)}, \ldots, \bar{\tilde{\Psi}}^{(p)}\right] \hat{\boldsymbol{\beta}}_{-i,\lambda} + \alpha_\lambda\right), \tag{5}$$

where $\hat{\boldsymbol{\beta}}_{-i,\lambda}$ is the LOO coefficient vector learned without sample $\mathbf{x}_i$.

4. **Evaluate predictions.** For some user-specified similarity metric $m$ and $k = 1, \ldots, p$, we define

$$MDI_k^+(\mathcal{S}, \mathcal{D}^*, \tilde{\Psi}, \mathcal{G}, m) := m(\mathbf{y}, \hat{\mathbf{y}}^{(k)}). \tag{6}$$

The resulting MDI+ scores are a $p$-vector denoting the importance of each feature to the trained tree-based model.

## B  Dataset Details

We selected datasets commonly used in machine learning benchmarks from the OpenML repository Vanschoren et al. (2013). We consistently used the six regression datasets and six classification datasets listed in Table 7 below. The *Geographic Origin of Music* dataset from the repository had many duplicate columns, so we removed duplicate columns to address redundancy. The *Ozone* dataset contained many highly correlated features, we therefore removed features with pairwise correlations greater than 0.95. Each dataset was split into training and testing sets. The experiments in Sections 4-6 allocate 67% of the data to the training set and 33% to the testing set. The experiments in Sections 7 and 8 require more test points and thus use a 50/50 split.

| | Name | Task ID | Features |
|---|---|---|---|
| **Regression** | Miami Housing | 361260 | 15 |
| | SARCOS | 361254 | 21 |
| | Puma Robot | 361259 | 32 |
| | Wave Energy | 361253 | 48 |
| | Geographic Origin of Music | 361243 | 72 |
| | Super Conductivity | 361242 | 81 |
| **Classification** | House 16H | 361063 | 16 |
| | Higgs | 361069 | 24 |
| | Pol | 361062 | 26 |
| | Ozone | 9978 | 47 |
| | Jannis | 361071 | 54 |
| | Spam | 43 | 57 |

Table 7: Real-world datasets used in the paper: regression (top panel), classification (bottom panel).

# C  Semi-Synthetic Feature Ranking Experiments

## C.1  Linear Response Function

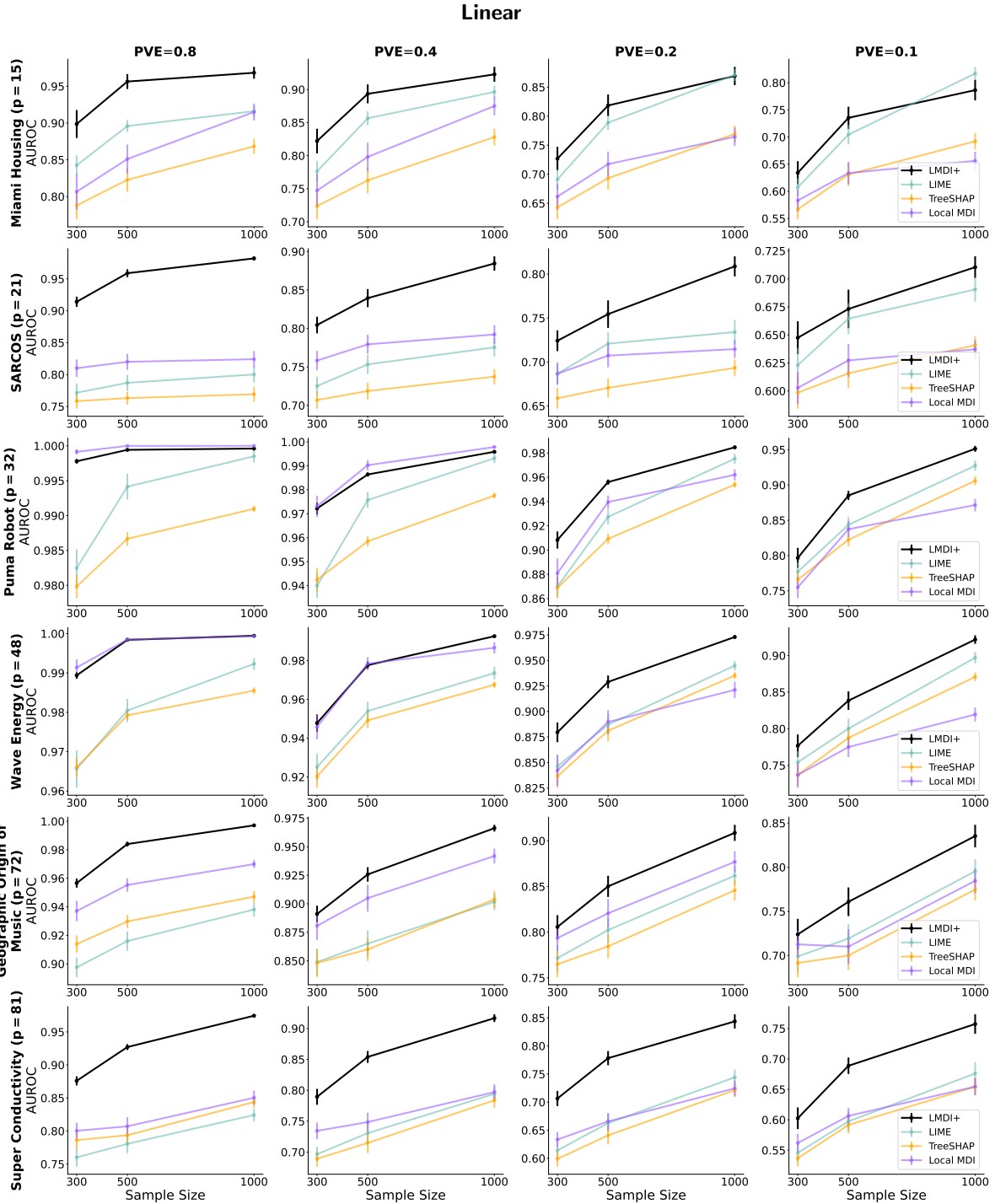

Figure 6: Across different sample sizes and signal-to-noise ratios in linear response function, LMDI+ demonstrates superior ability to distinguish signal features from non-signal features compared to other methods, with consistent trends observed across six regression datasets. Performance is reported as the average AUROC on test set samples and is averaged over 30 runs with different random sampling and the selection of signal features. Error bars show the standard error of the mean.

## C.2 Interaction Response Function

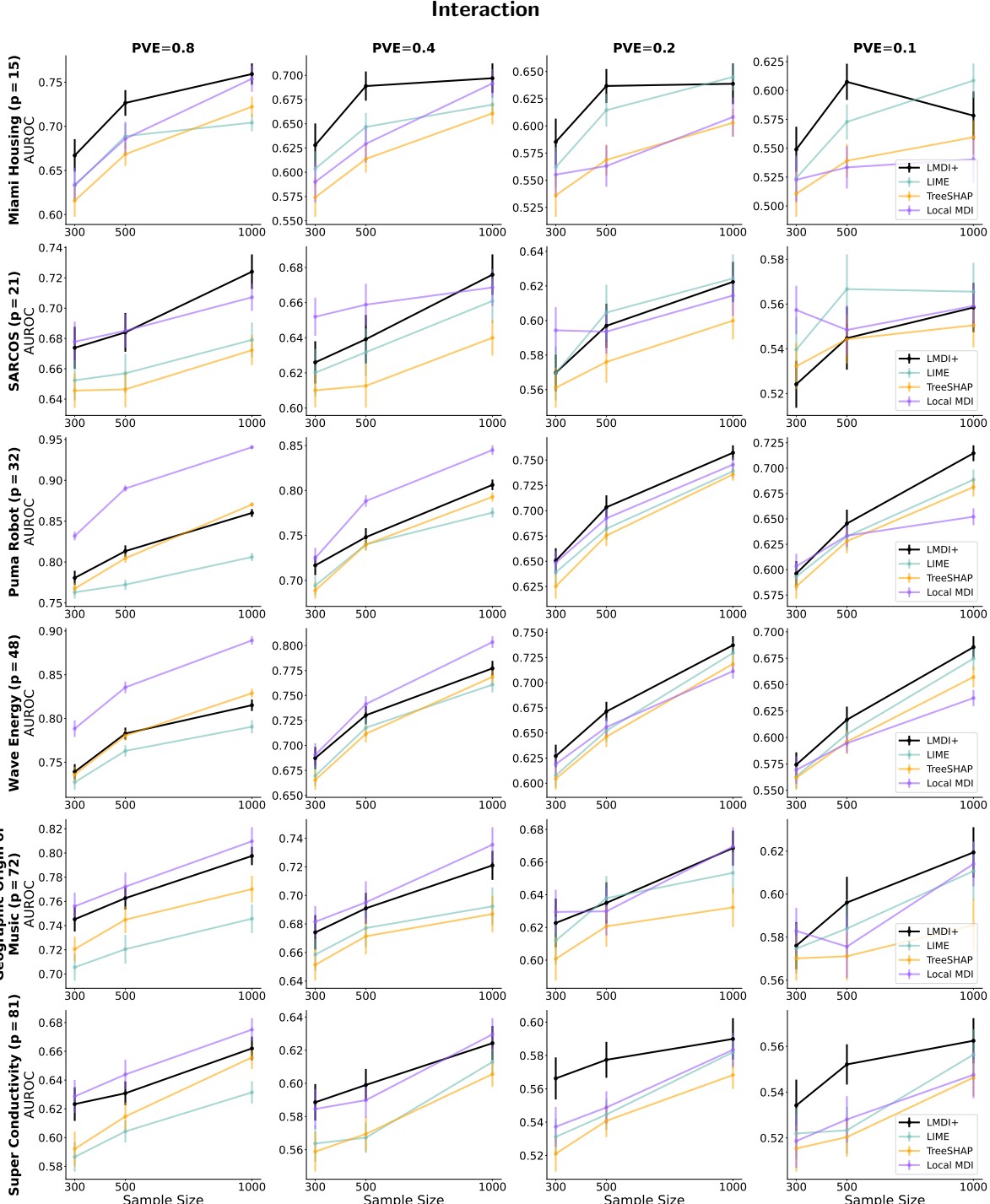

Figure 7: Across different sample sizes and signal-to-noise ratios in interaction response function, **LMDI+** demonstrates a competitive ability to distinguish signal features from non-signal features compared to other approaches, showing superior performance under low PVE and consistent trends across six regression datasets. Performance is reported as the average AUROC on test set samples and is averaged over 30 runs with different random sampling and the selection of signal features. Error bars show the standard error of the mean.

## C.3 Linear + LSS Response Function

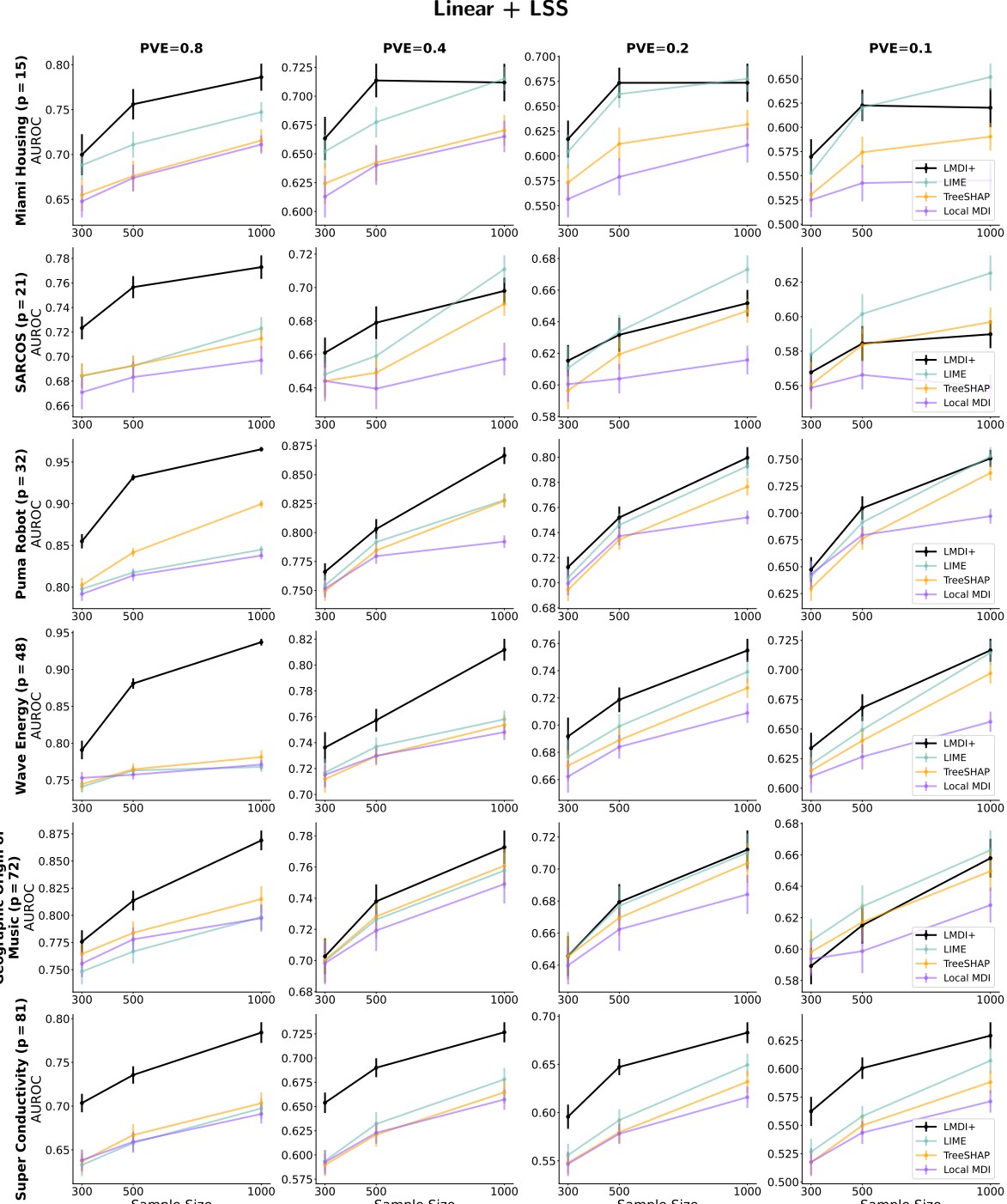

Figure 8: Across different sample sizes and signal-to-noise ratios in linear + LSS response function, LMDI+ demonstrates superior ability to distinguish signal features from non-signal features compared to other methods, with consistent trends observed across six regression datasets. Performance is reported as the average AUROC on test set samples and is averaged over 30 runs with different random sampling and the selection of signal features. Error bars show the standard error of the mean.

## C.4 Logistic Linear Response Function

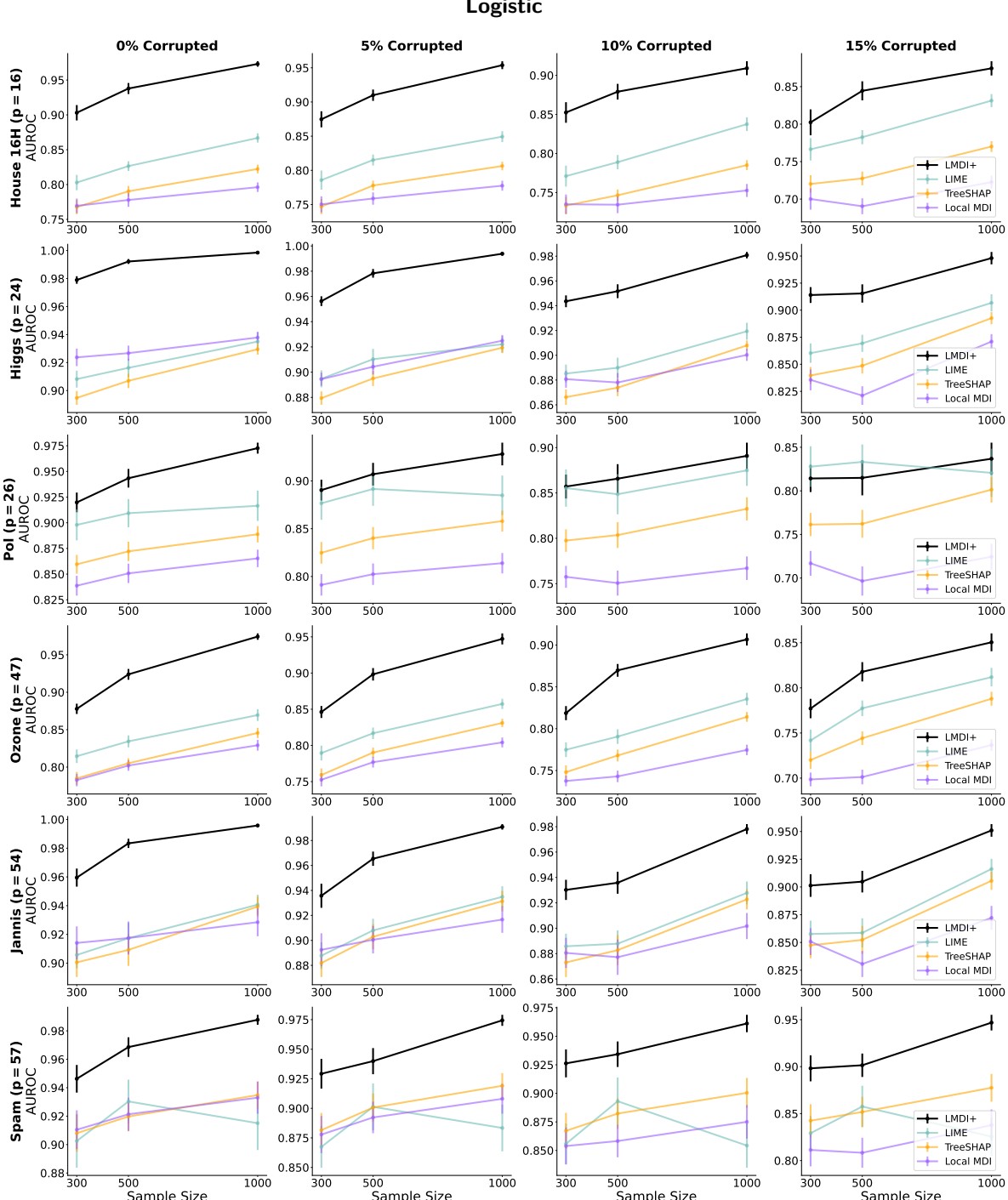

Figure 9: Across different sample sizes and signal-to-noise ratios in logistic response function, LMDI+ demonstrates superior ability to distinguish signal features from non-signal features compared to other methods, with consistent trends observed across six classification datasets. Performance is reported as the average AUROC on test set samples and is averaged over 30 runs with different random sampling and the selection of signal features. Error bars show the standard error of the mean.

## C.5 Logistic Interaction Response Function

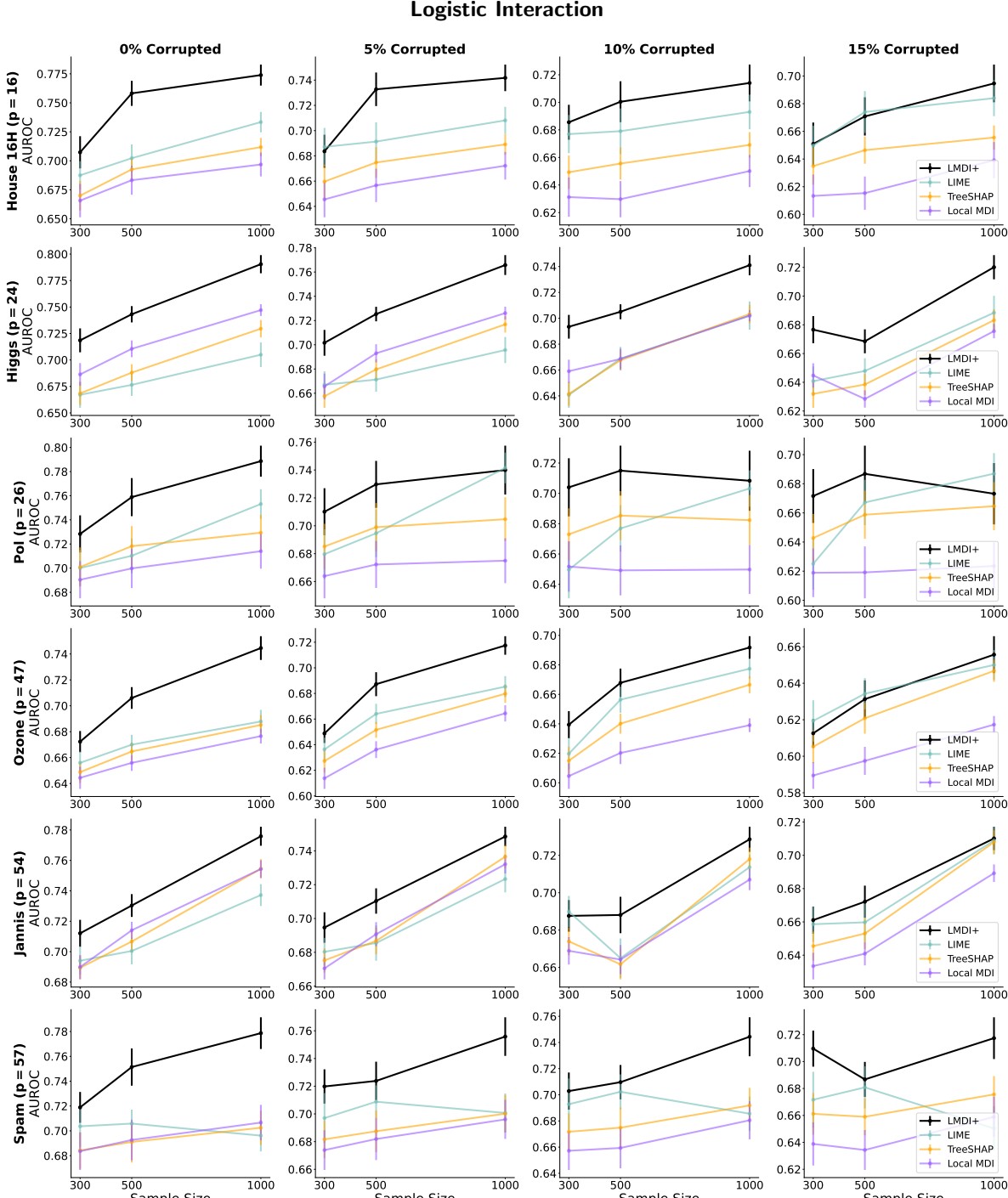

Figure 10: Across different sample sizes and signal-to-noise ratios in logistic interaction response function, LMDI+ demonstrates superior ability to distinguish signal features from non-signal features compared to other methods, with consistent trends observed across six classification datasets. Performance is reported as the average AUROC on test set samples and is averaged over 30 runs with different random sampling and the selection of signal features. Error bars show the standard error of the mean.

## C.6 Logistic Linear + LSS Response Function

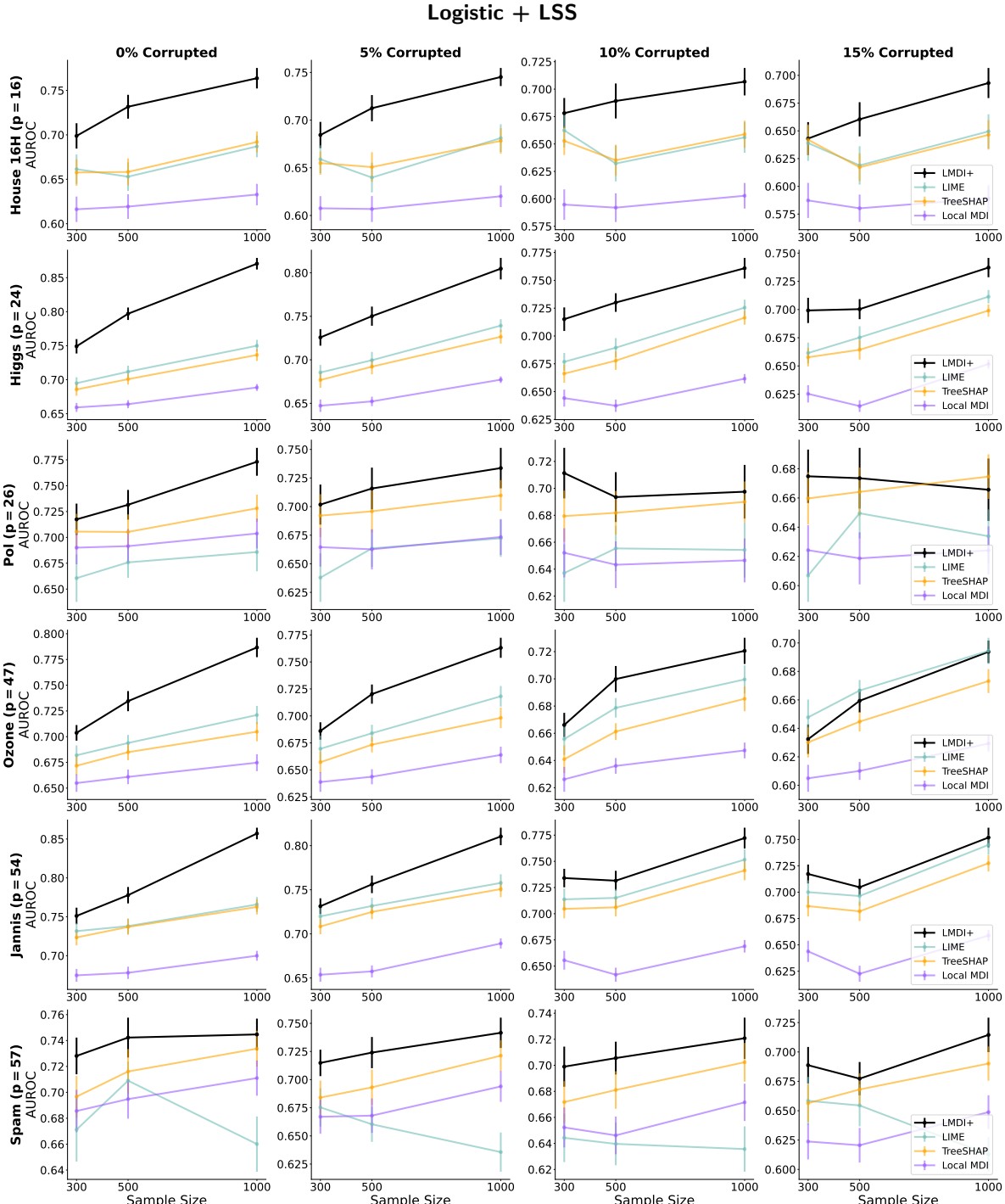

Figure 11: Across different sample sizes and signal-to-noise ratios in logistic linear + LSS response function, LMDI+ demonstrates superior ability to distinguish signal features from non-signal features compared to other methods, with consistent trends observed across six classification datasets. Performance is reported as the average AUROC on test set samples and is averaged over 30 runs with different random sampling and the selection of signal features. Error bars show the standard error of the mean.

# D   Real Data Experiments Results

## D.1   Experiments Results on Regression Datasets

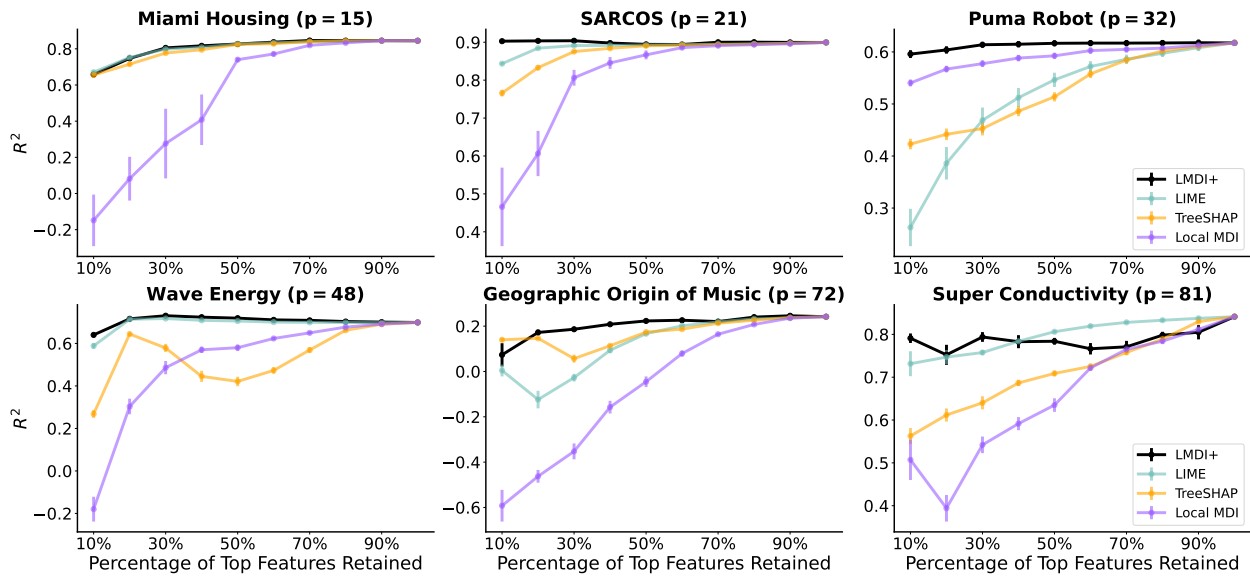

Figure 12: For regression benchmark datasets, LMDI+ achieves higher test $R^2$ after retraining on the masked training data selected by its feature ranking, demonstrating its ability to identify more signal features than baseline methods. Results are averaged over 20 runs with different random sampling and train-test splits. Error bars indicate the standard error of the mean.

## D.2   Experiments Results on Classification Datasets

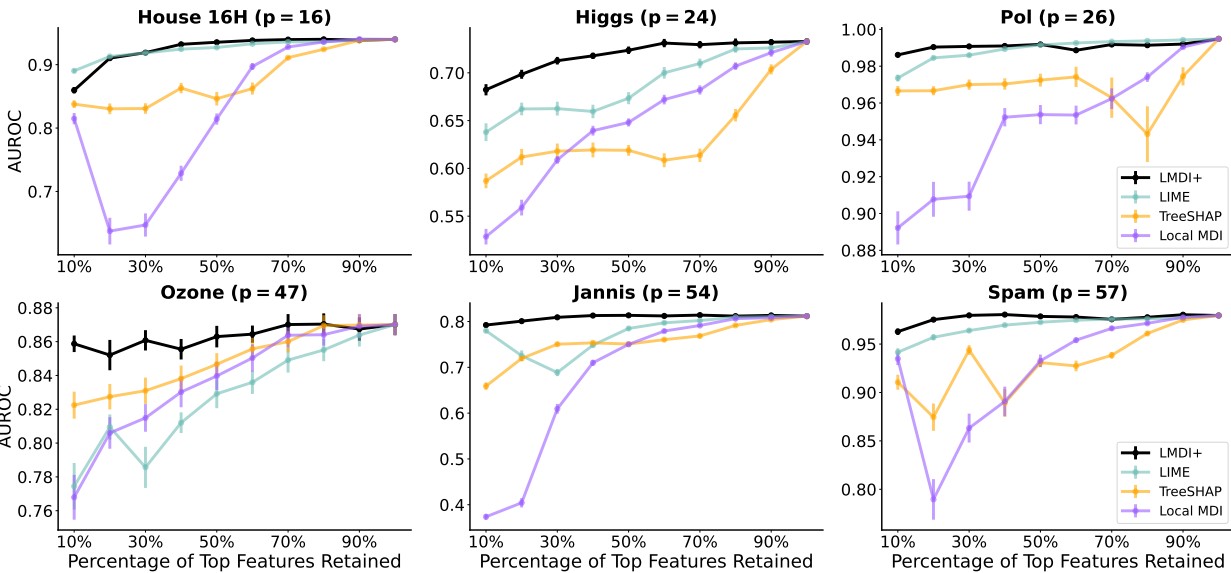

Figure 13: For classification benchmark datasets, LMDI+ achieves higher test AUROC after retraining on the masked training data selected by its feature ranking, demonstrating its ability to identify more signal features than baseline methods. Results are averaged over 20 runs with different random sampling and train-test splits. Error bars indicate the standard error of the mean.

### D.3 Additional Real Data Experiments on Full Datasets

To demonstrate that our findings hold beyond the downsampled setting, we reproduce the real data experiments on the full datasets, excluding LIME due to its prohibitive runtime on large datasets. LMDI+ continues to outperform TreeSHAP and Local MDI across datasets and feature retention levels, demonstrating that our findings hold beyond the downsampled setting.

| Method | Classification (AUROC ↑) | | Regression ($R^2$ ↑) | | | |
|---|---|---|---|---|---|---|
| | Pol (N=10,082) | House 16H (N=13,488) | Miami Housing (N=13,932) | Super Conductivity (N=21,263) | SARCOS (N=48,933) | Wave Energy (N=72,000) |
| *Keep Top 10%* | | | | | | |
| LMDI+ | **0.9922** | **0.8594** | **0.6491** | **0.8268** | **0.9445** | **0.7171** |
| TreeSHAP | 0.9715 | 0.6153 | 0.6127 | 0.5354 | 0.7274 | -0.1937 |
| Local MDI | 0.7520 | 0.6873 | -0.6994 | 0.5368 | 0.1567 | -0.9649 |
| *Keep Top 20%* | | | | | | |
| LMDI+ | **0.9959** | **0.9253** | **0.7650** | **0.8151** | **0.9562** | **0.8250** |
| TreeSHAP | 0.9784 | 0.8149 | 0.7625 | 0.6630 | 0.8855 | 0.7360 |
| Local MDI | 0.9038 | 0.5853 | -0.2378 | 0.4545 | 0.6484 | 0.4935 |
| *Keep Top 30%* | | | | | | |
| LMDI+ | **0.9931** | **0.9310** | **0.8603** | **0.7581** | **0.9643** | **0.8521** |
| TreeSHAP | 0.9845 | 0.7659 | 0.8080 | 0.6700 | 0.9321 | 0.7630 |
| Local MDI | 0.9396 | 0.6042 | -0.2883 | 0.6564 | 0.8610 | 0.4863 |

Table 8: Real Data Experiments using full datasets, excluding LIME due to runtime constraints. Bold indicates the best performance for each dataset and keep ratio.

# E   Stability Experiments Results

## E.1   Stability Results on Regression Datasets

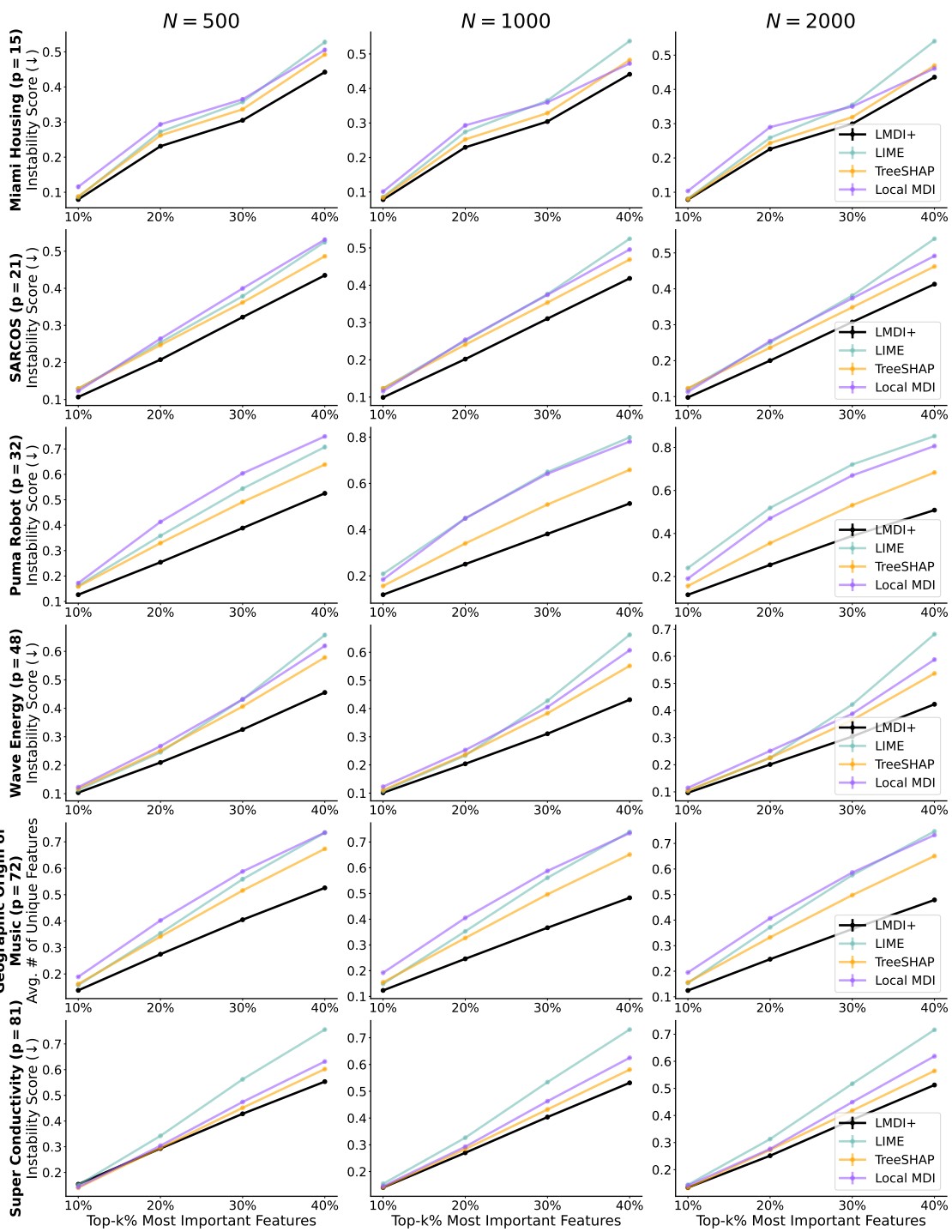

Figure 14: Across all regression datasets, LMDI+ yields more stable feature rankings as it identifies a smaller number of unique features when selecting the top important features across different random forest fits on the same data. Performance is reported as the average on the entire set and is averaged over 15 runs with different random sampling and train-test splits. Error bars showing the standard error of the mean.

## E.2 Stability Results on Classification Datasets

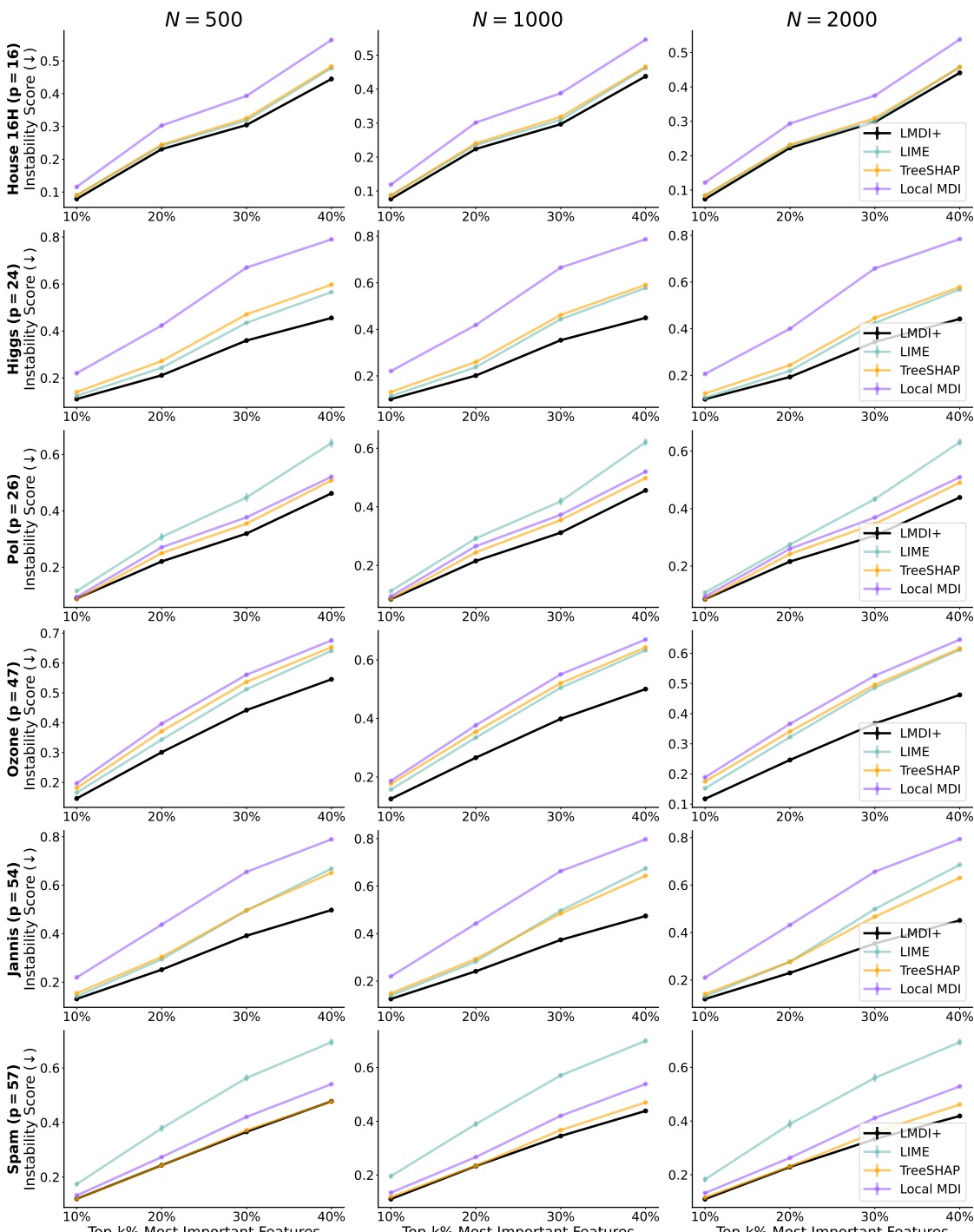

Figure 15: Across all classification datasets, LMDI+ yields more stable feature rankings as it identifies a smaller number of unique features when selecting the top important features across different random forest fits on the same data. Performance is reported as the average on the entire set and is averaged over 15 runs with different random sampling and train-test splits. Error bars showing the standard error of the mean.

# F   Ablation Analysis

Here we present the complete results of the ablation study, where we evaluate LMDI+ under additional ensemble configurations. Specifically, we vary the number of estimators (`n_estimators` $\in \{10, 50\}$) and the minimum number of samples per leaf (`min_samples_leaf` $\in \{5, 10\}$), and additionally evaluate performance with gradient boosting ensembles. Table 9 reports the corresponding results.

**Average Rank** ($\downarrow$)

| | | Percent of Features Retained | | | |
| | | Feature Selection | | Stability | |
| **Ensemble** | **Method** | 10% | 20% | 10% | 20% |
|---|---|---|---|---|---|
| RF (`n_estimators=10`) | LMDI+ | **1.58** | **1.25** | **1.17** | **1.00** |
| | LIME | 2.42 | 2.58 | 2.92 | 2.83 |
| | TreeSHAP | 2.17 | 2.42 | 2.42 | 2.42 |
| | Local MDI | 3.83 | 3.75 | 3.50 | 3.75 |
| RF (`n_estimators=50`) | LMDI+ | **1.33** | **1.17** | **1.17** | **1.00** |
| | LIME | 2.17 | 2.33 | 2.92 | 2.83 |
| | TreeSHAP | 2.75 | 2.75 | 2.42 | 2.42 |
| | Local MDI | 3.75 | 3.75 | 3.50 | 3.75 |
| RF (`min_sample_leaf=5`) | LMDI+ | **1.25** | **1.08** | **1.17** | **1.00** |
| | LIME | 2.50 | 2.33 | 2.92 | 2.83 |
| | TreeSHAP | 2.58 | 2.92 | 2.42 | 2.42 |
| | Local MDI | 3.67 | 3.67 | 3.58 | 3.25 |
| RF (`min_sample_leaf=10`) | LMDI+ | **1.33** | **1.17** | **1.17** | **1.00** |
| | LIME | 2.42 | 2.25 | 2.92 | 2.83 |
| | TreeSHAP | 2.58 | 2.75 | 2.42 | 2.42 |
| | Local MDI | 3.67 | 3.83 | 3.50 | 3.25 |
| Gradient Boosting | LMDI+ | **1.08** | **1.08** | **1.17** | **1.00** |
| | LIME | 2.33 | 2.25 | 2.75 | 2.75 |
| | TreeSHAP | 2.58 | 2.67 | 2.25 | 2.25 |

Table 9: Ablation study for LMDI+ across different ensembles. Results show the average rank across all datasets, where lower is better.

# G Counterfactuals on Local Feature Importance Prioritize Signal Features

## G.1 Experiment Setup

For $i = 1, \ldots, 2000$, we take $\mathbf{x}_i \sim \mathcal{N}_{10}(0, I_{10})$. We set coefficients $\boldsymbol{\beta} = [5, 4, 3, 2, 1, 0, 0, 0, 0, 0]$ such that $X_1, \ldots, X_5$ are signal features with $X_1$ being the strongest signal and $X_6, \ldots, X_{10}$ are noise features. We then generate labels $\mathbf{y}$ by taking $\mathtt{sign}(\mathbf{X}\boldsymbol{\beta} + \epsilon)$, with $\epsilon = 0.1$. We note this is a high signal-to-noise ratio: a logistic regression model attains 99% accuracy on held-out data.

We fit a random forest to half of the data and compute local feature importances on both the training and held-out test data. We compute counterfactuals both by using the observed data and local feature importances as described in Section 7.

## G.2 Results

In Figure 16 we show the distribution of coordinate-wise distances to the nearest counterfactual training observation. Note that features 1-5 are signal features with decreasing levels of signal. Thus, we want our counterfactual to be closest with respect to $X_1$, then $X_2$, and so on. We see that the counterfactuals found using the observed data tend to be closer with respect to the noise features $X_6, \ldots, X_{10}$. Meanwhile, the local feature importance methods find counterfactuals that are closest with respect to signal features, with the distance increasing as signal lessens.

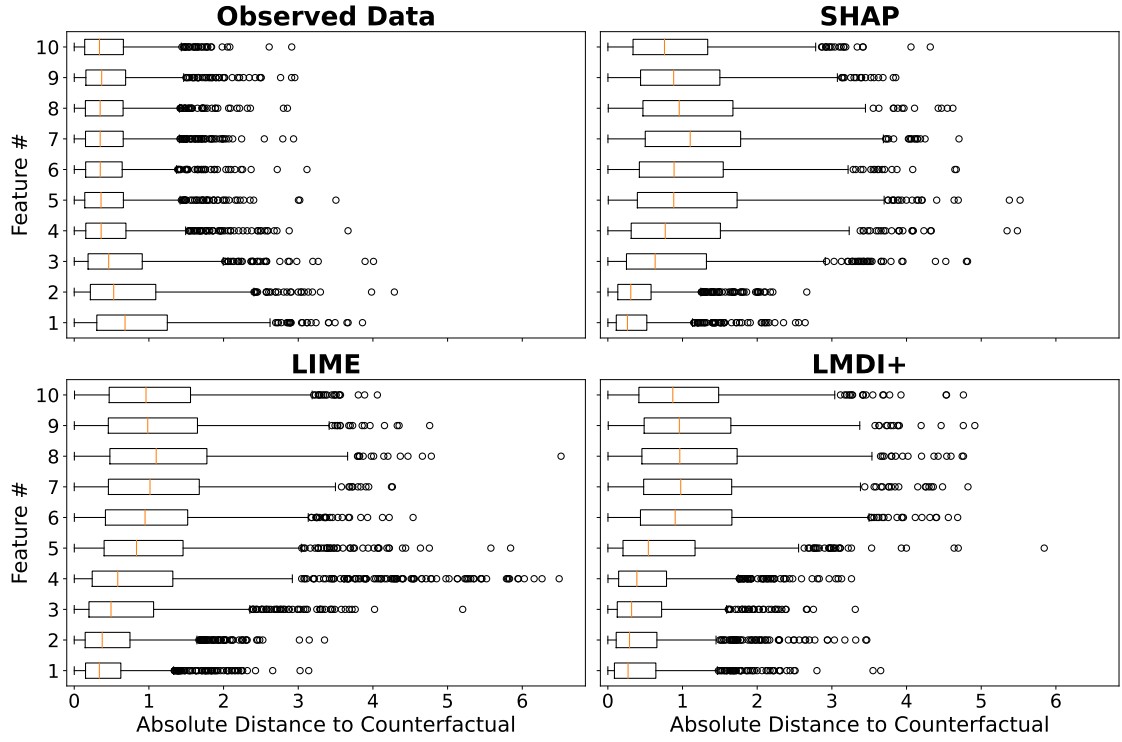

Figure 16: Distribution of coordinate-wise distances between an observation and its counterfactual. When using local feature importance space to find the closest counterfactual, the detected explanation is more similar with respect to signal features. Using the observed data, which is the typical approach, instead results in counterfactual explanations that are more similar with respect to noise features.

## H    Exploratory Subgroup Analysis

In this section, we investigate the patterns within the subgroups formed by LMDI+. We select $k = 4$ clusters using the stability method described in Algorithm 1 of Ben-Hur et al. (2001). We begin by visualizing the cluster on a map of Miami, which can be seen in Figure 17. We witness a clear geographic trend, with cluster 4 containing the houses closest to the city center, cluster 2 containing the houses in wealthy suburbs such as Coral Gables, Pinecrest, and Palmetto Bay, and clusters 1 and 3 containing the houses in the outlying suburbs of Miami-Dade County.

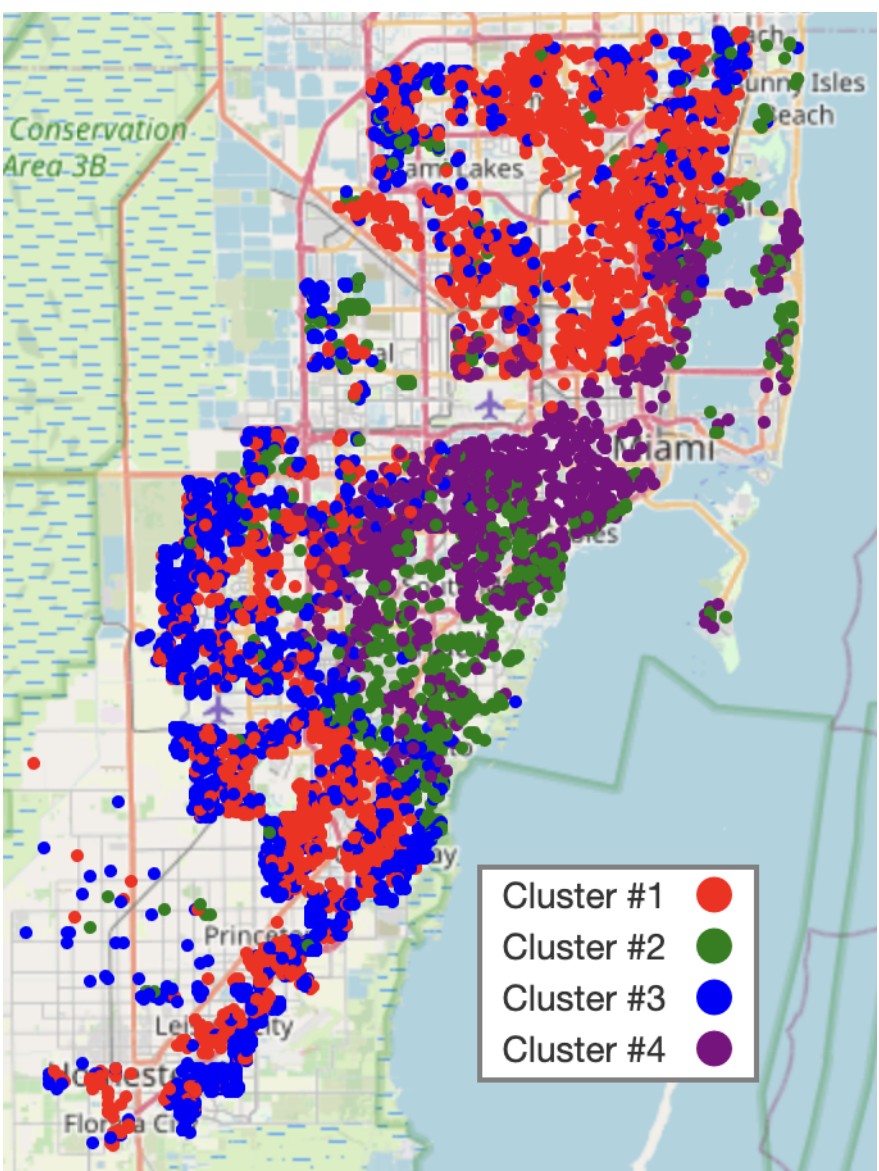

Figure 17: Homes in Miami-Dade County, location determined by latitude and longitude. We witness clear geographic trends, with Cluster 1 and 3 comprising the outer suburbs, Cluster 2 comprising of the wealthier coastal suburbs, and Cluster 4 containing the homes closer to the city center.

The left side of Figure 18 allows us to examine living area, a key driver of house prices. We observe a clear separation between clusters 1 and 3, implying that subgroup 1 contains the smaller suburban homes than subgroup 3, allowing us to differentiate between them. Investigating home price by cluster, which can be seen

on the right side of Figure 18, completes the story. We see that homes in cluster 1 are the least expensive, likely due to a combination of their size and distance from the city center. Despite their similar location to cluster 1, homes in cluster 3 consistently sell for higher prices due to their larger nature. Finally, we see that homes in cluster 4 are valuable due to their proximity to the city center, while homes in cluster 2 are valuable due to their large size.

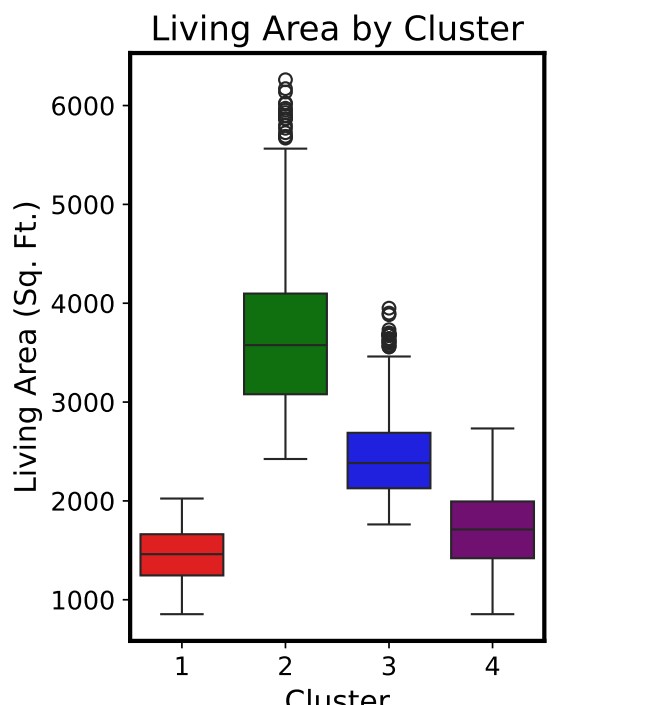 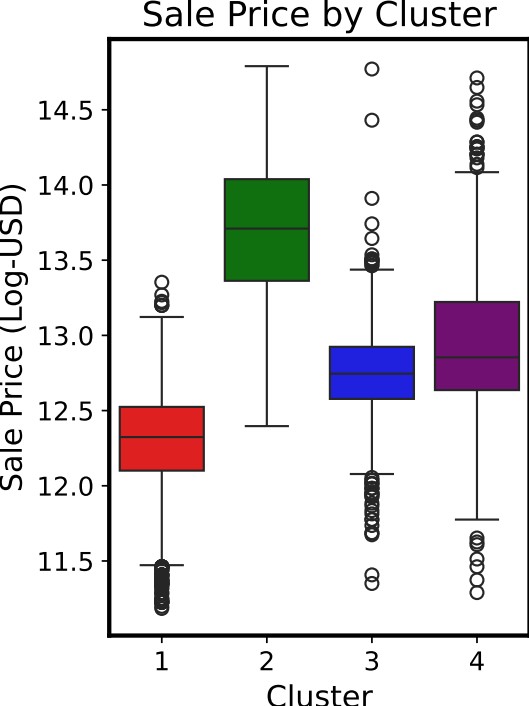

Figure 18: We observe that the distributions of living area and home price differ drastically between clusters, indicating key characteristics of each group.

