# OpenReview forum: "Local MDI+: Local Feature Importances for Tree-Based Models"
_TMLR — Accepted by TMLR_

### Review · Reviewer_Fut2 · 2026-02-18

**Summary Of Contributions:**

### Summary
This paper introduces a novel method for estimating Local Feature Importance called LMDI+. This new method is based on the Mean Decrease in Impurity for tree-based architectures, referred to as MDI+. MDI+ employs least squares to determine global feature importance. The paper begins by outlining a sequence of methods (MDI, Local MDI, and MDI+) along with their motivations and limitations, which effectively sets the stage for the proposed method, LMDI+. Experiments demonstrate that the proposal achieves higher performance than prior approaches and greater robustness across both synthetic and real data benchmarks. Additionally, the paper highlights improved stability in feature-importance ranking and includes an ablation analysis for the components of the proposed method.

### Strengths

- The paper presents clearly the motivations behind each decision. The authors do a good job of explaining the state of the art and how the various previous methods were developed, providing a narrative that is understandable to readers who are not necessarily familiar with the details of the research area, and also clear about how the proposal was developed.
- The method presented is simple and intuitive. Building on previous work, LMDI+ is presented, which improves not only the performance of tree-based methods but also their stability.
- The results presented are comprehensive. The method is evaluated on synthetic datasets to assess feature identification and on real benchmarks to assess whether the selected features are effectively helpful for training.

### Weaknesses
- What has been presented is only an incremental and natural step forward from previous work. Despite this, the authors effectively explain and motivate their ideas and proposals.
- There are sections in the appendix that should be added to the main paper.
    - Details about the experimental setup should be in a subsection at the beginning of Section 4.
    - Details about time and computational complexity should be added in the experiments section. This subsection is even more relevant, since in Section 4.3 it is mentioned that: “For computational efficiency, datasets with more than 2000 samples were downsampled to 2000”
    - How does the number of features affect the method? With a smaller number of features, does the performance difference between LFI methods decrease? I ask this because it is mentioned that the datasets were selected based on the number of features.

### Questions
Given my background outside this specific research area, I have several questions that may or may not be relevant to this work.
- What is the motivation for using a local feature selection or importance? I understand the motivation for selecting features at the global level, as these help identify features that improve performance. But at the local (sample) level, why is it relevant?
    - If a feature is relevant to a sample but not to the dataset, that could indicate overfitting. Or not?
- In Section 4.2. What is the intuition behind the model selecting the Non-Signal to be more important than both the Signal and the Cor. Non-Signal?
    - One would expect that as the correlation increases, the rankings of both Signal and Cor. Non-signal would be more similar, as the relevance of one should be more closely correlated with the importance of the other. Why is this intuition not correct in LMDI+? It seems that the intuition would make sense, but the correlation = 0.99 went in a different direction (Figure 2)
- In Section 5. I understand the motivation and the experiments, but the results do not align with those presented in Section 4 of this paper.
    - The selection can be more stable in terms of which features the methods select, but this does not affect the model's performance (or not drastically), as shown in Figure 3. It may be important to use a more stable method when this selection is associated with performance instability. However, we observe that this instability does not necessarily affect performance.
- A good experiment would show that, as stability increases, the standard deviation of performance decreases.
    - What is the intuition behind the improvement in some cases and not in others? For example. When using feature selection, the Rank between 2 and 3 remains the same at 10%, but decreases to 20% with 20%. Is there an intuition behind this, or is it the std?

**Audience:**

Yes

**Audience Explanation:**

Although I am not an expert in Local Feature Importance, nor do I use these methods regularly, I understand that such methods are relevant for many scenarios in which variables can create spurious correlations, and that importance selection can help reduce overfitting. This is especially relevant in scenarios where interpretability is important, such as in healthcare benchmarks.

**Broader Impact Concerns:**

There is no Broader Impact Statement. However, one potential impact of these types of methods could be on the robustness of the selection in out-of-distribution settings. For example, could the selection of feature importance deliver a spurious combination which could help achieve better performance in the current distribution but could ultimately harm the generalisation to elements that may be slightly outside the distribution used in training and testing? Are there any studies that perform a robustness analysis on this area?

**Claims And Evidence:**

Yes

**Claims Explanation:**

The authors present a well-motivated set of experiments. Starting from a synthetic dataset, the paper shows that the feature importance method outperforms prior methods across varying signal-to-noise ratios and correlation levels. The paper then shows higher performance on real data benchmarks across different feature retention percentages, thereby complementing the previous results. The paper then presents a comprehensive ablation analysis of the method's components.

**Requested Changes:**

- Moving the experimental details to the main paper, along with a clearer explanation of the hyperparameter selection. Adding a brief analysis of the tree hyperparameters can help mitigate potential doubts.
- Another suggestion is to move the runtime analysis (Appendix J) to the main paper.
- Provide a clearer connection to the results presented in the Appendix. Most of the time, there is one small reference without a clear explanation for what is in that Section, which suggests that what is in the Appendix is not relevant.

---

> ### Author Response · Authors · 2026-03-12
> **Response to Reviewer Fut2 [Part 1]**
>
> We thank the reviewer for the thorough and constructive review. We have addressed your concerns below and revised the paper accordingly. The updated sections are highlighted in blue.
>
> **Questions:**
>
> **Q: What is the motivation for using a local feature selection or importance? If a feature is relevant to a sample but not to the dataset, that could indicate overfitting. Or not?**
>
> A: Global feature importances measure which features the model relies on across the entire dataset. These global values are informative in telling us the “average” behavior of the model, but local importances are able to go one level deeper in explaining the model predictions for a particular sample. For example, let’s say a medical practitioner is using a tree-based model to help with the diagnosis of some condition. If a particular patient would like an explanation of why they were diagnosed, what drives the model on average might not actually be how the model reached its conclusion. If this patient has a rare underlying condition that happens to be the primary driver of their diagnosis, it may have very low global importance on average (since it is rare in the population) but be the major reason the model reached its conclusion for their particular case.
>
> This is the general strength of local importance. In practice, the data often consists of multiple subgroups, where the subgroups each have a heterogeneous relationship with the outcome. These differences can be accurately described by local feature importances in a way that is not possible with one set of global importance scores (Section 8).
>
> **Q: In Section 4.2. What is the intuition behind the model selecting the Non-Signal to be more important than both the Signal and the Cor. Non-Signal? ... Why is this intuition not correct in LMDI+?**
>
> A: There are two separate questions here. We first explain why other local feature importance methods (LIME, TreeSHAP, Local MDI) tend to rank the non-correlated non-signal features as most important in the high-correlation regime ($\rho=0.99$), and then explain why LMDI+ handles this better.
>
> Recall the setup of Section 4.2, where we have three feature groups: signal features (correlation $\rho$ with each other), non-signal features that have correlation $\rho$ with the signal features, and non-signal features that are independent of all other features. When $\rho$ is very high, this means that the first two groups (signal and correlated non-signal) contain nearly identical information, so any feature from either group can yield splits with roughly similar gain. As a result, different trees select different features from this correlated group, spreading the importance of the true signal across all correlated features. This artificially deflates the importance of the true signal features while inflating that of the correlated non-signal features, leaving the independent non-signal features ranked highest.
>
> LMDI+ is better equipped to handle the high-correlation setting due to (1) the use of out-of-bag samples, and (2) the use of a regularized GLM rather than the traditional linear model. Whereas traditional tree-based feature importances (e.g. MDI) consider only in-bag samples (the samples used in fitting the tree), LMDI+ uses both in-bag and out-of-bag samples, helping fight overfitting. Using a regularized GLM also allows LMDI+ to inherit the benefits of these models, since the penalization allows for a more graceful handling of importance in unstable/correlated feature space compared to OLS.

---

> > ### Author Response · Authors · 2026-03-12
> > **Response to Reviewer Fut2 [Part 2]**
> >
> > **Q: In Section 5. I understand the motivation and the experiments, but the results do not align with those presented in Section 4 of this paper… A good experiment would show that, as stability increases, the standard deviation of performance decreases.**
> >
> > A: Thank you for the thoughtful suggestion. We have conducted the requested experiment to quantify performance instability across random seeds. Concretely, for each method we keep the same experimental setup and train five RF models, each initialized with a different random seed. We then compute the standard deviation of feature-selection performance (Section 4.3) across these seeds for varying top-feature keep ratios. Table 1 reports the results averaged across datasets.
> > The results show that LMDI+ consistently yields a lower standard deviation than the baselines, indicating that its improved selection stability is accompanied by more stable feature-selection performance across runs.
> > | Method | Keep 10% | Keep 20% | Keep 30% | Keep 40% |
> > |---|---:|---:|---:|---:|
> > | LMDI+ | **0.0115** | **0.0082** | **0.0050** | **0.0046** |
> > | LIME | 0.0220 | 0.0175 | 0.0133 | 0.0096 |
> > | TreeSHAP | 0.0125 | 0.0094 | 0.0058 | 0.0048 |
> > | Local MDI | 0.0163 | 0.0103 | 0.0063 | 0.0058 |
> >
> > **Table 1.**  Standard deviation of feature-selection performance across five random seeds, averaged across datasets and reported for varying top-feature keep ratios per sample. Bold indicates the lowest standard deviation for each keep ratio. LMDI+ consistently achieves the lowest standard deviation across keep ratios, indicating more stable performance across runs.
> >
> >
> > **Q: What is the intuition behind the improvement in some cases and not in others? For example. When using feature selection, the Rank between 2 and 3 remains the same at 10%, but decreases to 20% with 20%. Is there an intuition behind this, or is it the std?**
> >
> > A: When retaining only 10% of features, the top-decile features tend to have well-separated importance scores, so both Variant 2 and Variant 3 identify the same high-signal subset regardless of whether raw features are included. But we do observe that Variant 3 is more stable than Variant 2, consistent with the benefit of adding raw features. When retaining 20%, the method must also rank borderline features whose importance scores are less separated and noisier. Here, the richer feature representation in Variant 3 provides better discrimination among these mid-tier candidates, leading to a more reliable ordering and the observed improvement in average rank.
> >
> >
> > **Q: How does the number of features affect the method? With a smaller number of features, does the performance difference between LFI methods decrease?**
> >
> > A: Yes, because with very few features, most LFI methods often yield similar rankings, making differences harder to observe. We therefore selected datasets with a larger number of features to better stress-test the methods and make performance gaps more discernible. Our benchmark still spans a range of feature counts (15–81).

---

> ### Author Response · Authors · 2026-03-12
> **Response to Reviewer Fut2 [Part 3]**
>
> **Q: One potential impact of these types of methods could be on the robustness of the selection in out-of-distribution settings … Are there any studies that perform a robustness analysis on this area?**
>
> A: We thank the reviewer for raising this important point. While we do not conduct explicit out-of-distribution experiments, we argue that LMDI+'s design choices position it more favorably than baseline methods on this dimension for several reasons. First, the use of regularized GLMs explicitly penalizes reliance on spuriously correlated features, reducing the risk of selecting importance combinations that are artifacts of the training distribution. Second, the use of OOB samples to evaluate feature importance already provides a layer of protection against in-sample overfitting.
>
> We also note that evaluating LFI methods under distribution shift is not straightforward in practice: since LFI methods explain the predictions of a specific trained model, assessing their behavior under distribution shift would typically require retraining the model on the shifted distribution, at which point the explanation target itself has changed. As a result, the existing LFI literature has largely not adopted OOD robustness as a standard evaluation criterion, instead focusing on predictivity and stability within a fixed distribution [1-4], which is the evaluation framework we follow in Sections 4 and 5. We agree this is nonetheless an important and underexplored direction for future work, and we will add an explicit discussion of this limitation to the paper.
>
> [1] Muhammad Rehman Zafar and Naimul Mefraz Khan. Dlime: A deterministic local interpretable model-agnostic explanations approach for computer-aided diagnosis systems. arXiv:1906.10263, 2019.
>
> [2] Xuanxiang Huang and Joao Marques-Silva. On the failings of shapley values for explainability. International Journal of Approximate Reasoning, 171:109112, 2024. ISSN 0888-613X. doi: https://doi.org/10.1016/j.ijar.2023.109112. URL https://www.sciencedirect.com/
> science/article/pii/S0888613X23002438. Synergies between Machine Learning and
> Reasoning.
>
> [3] Yujia Zhang, Kuangyan Song, Yiming Sun, Sarah Tan, and Madeleine Udell. " why should
> you trust my explanation?" understanding uncertainty in lime explanations. arXiv:1904.12991, 2019.
>
> [4] David Alvarez-Melis and Tommi S Jaakkola. On the robustness of interpretability methods.
> arXiv:1806.08049, 2018.
>
>
> **Requested Changes:**
>
> A: Thanks for the helpful feedback. We have revised the manuscript by moving the experimental details and the hyperparameter selection procedure (previously Appendix B) into the main paper (Section 4), and by moving the runtime analysis (previously Appendix J) into the main paper (Section 9). We have also revised the Appendix references throughout to briefly summarize what each referenced appendix section contains.
>
> We hope that our clarifications, revisions, and additional results provide stronger evidence for the effectiveness of LMDI+. We welcome any further suggestions you may have.

---

### Review · Reviewer_LD77 · 2026-02-25

**Summary Of Contributions:**

The paper proposes a new method for local (sample-specific) feature importance analysis for trees and tree ensembles. The proposed method extends and modifies MDI+, endowing it with the capability to quantify feature importance for each sample while retaining its strengths. The paper reports on experiments that demonstrate, at least on small benchmark datasets, that the proposed method improves upon existing local feature importance techniques. The results suggest that the method generates a better ranking of features, is less susceptible to high correlation effects, and provides more robust selection. An ablation study establishes that each of the three components of the proposed method contribute to the improved performance.

**Audience:**

Yes

**Audience Explanation:**

The paper addresses feature importance analysis for trees and tree ensembles, which are very commonly used for tabular data. There are a large number of practitioners who read TMLR and use such models; they would be interested in learning about a new and improved method for performing feature importance analysis.

**Claims And Evidence:**

Yes

**Claims Explanation:**

The paper makes the following claims:

(1)	The paper proposes a novel extension of the MDI+ framework that quantifies feature importance for each sample.

(2)	The proposed method, LMDI+, outperforms existing baselines across 12 benchmark datasets, in term of identifying predictive features (yielding an average predictive performance improvement of 10 percent when using only the selected features)

(3)	The technique exhibits greater stability, providing more consistent rankings across different model fits.

(4)	Ablation studies demonstrate the value of each component of LMDI+ and illustrate that the improvements extend to gradient boosting models

(5)	LMDI+ can be used to identify counterfactuals for the classification benchmarks and identify subgroups for a housing dataset.

Claim 1 is supported by the methodology section where a new method is presented. Claim 2 is supported by experimental results on the 12 datasets that are analyzed. One issue with this claim is that all datasets are reduced to at most 2000 samples. As a result, the claim is not really established for the original benchmarks. Claims 3-5 are supported by experimental results and a careful ablation analysis.

**Requested Changes:**

Overall the paper is well written and provides sufficient support for the claims. While the methodological changes for the proposed method are not dramatic, the experimental results demonstrate a meaningful performance improvement.

(1)	Correlation analysis
While the results in Section 4.2 provide some evidence that correlation is better handled by LMDI+, they explore only one setting (where features are all either correlated, with the same correlation value, or not correlated at all).

Please add a specification of the number of signal features (and how many are correlated and not). This will help the reader to understand what an ideal average rank value is.

The paper does not explain why there is a dip in LMDI+ from 0.9 to 0.99. Given the general trend of all other curves and LMDI+ from 0.5 to 0.9, this definitely needs an explanation.

Since the paper stresses that LMDI+ handles very high correlations better than the other methods, there should be more values investigated between 0.85 and 0.99.

(2)	Computational complexity

Section 9 provides a brief discussion of computational complexity and there is a more detailed runtime analysis in an appendix.
The truncation of the datasets to 2000 samples is somewhat concerning. This is by no means a large dataset.

Could the authors please comment on what happens with larger datasets (tens of thousands of samples)? Are the results similar to what is reported here or are there qualitative differences?

---

> ### Author Response · Authors · 2026-03-12
> **Response to Reviewer LD77 [Part 1]**
>
> We thank the reviewer for their thoughtful feedback. We have addressed your concerns below and revised the paper accordingly. The updated sections are highlighted in blue.
>
> **Q: One issue with this claim is that all datasets are reduced to at most 2000 samples. As a result, the claim is not really established for the original benchmarks...Could the authors please comment on what happens with larger datasets (tens of thousands of samples)?**
>
> A: We thank the reviewer for raising this important point. The downsampling to 2000 samples was primarily done to enable fair comparison with LIME, which as shown in Section 9 runs approximately 5x-10x slower than LMDI+ and becomes computationally prohibitive on larger datasets. As requested, we have now rerun the real data experiments from Section 4.3 on the full datasets for the larger benchmarks, excluding LIME due to its runtime constraints. Results are reported below. LMDI+ continues to outperform TreeSHAP and Local MDI across datasets and feature retention levels, demonstrating that our findings hold beyond the downsampled setting. We have revised Section 4.3 to clarify the rationale for downsampling and included the full dataset experiments in Appendix D.
>
> We have also moved the runtime analysis from the Appendix into Section 9 and note that our implementation parallelizes computation across trees, allowing LMDI+ to run consistently faster than LIME and with runtime comparable to TreeSHAP.
>
> | Method | Pol (N=10082, AUROC) | House 16H (N=13488, AUROC) | Miami Housing (N=13932, R²) | Super Conductivity (N=21263, R²) | SARCOS (N=48933, R²) | Wave Energy (N=72000, R²) |
> |:---|:---:|:---:|:---:|:---:|:---:|:---:|
> | **Keep Top 10%** | | | | | | |
> | LMDI+ | **0.9922** | **0.8594** | **0.6491** | **0.8268** | **0.9445** | **0.7171** |
> | TreeSHAP | 0.9715 | 0.6153 | 0.6127 | 0.5354 | 0.7274 | -0.1937 |
> | Local MDI | 0.7520 | 0.6873 | -0.6994 | 0.5368 | 0.1567 | -0.9649 |
> | **Keep Top 20%** | | | | | | |
> | LMDI+ | **0.9959** | **0.9253** | **0.7650** | **0.8151** | **0.9562** | **0.8250** |
> | TreeSHAP | 0.9784 | 0.8149 | 0.7625 | 0.6630 | 0.8855 | 0.7360 |
> | Local MDI | 0.9038 | 0.5853 | -0.2378 | 0.4545 | 0.6484 | 0.4935 |
> | **Keep Top 30%** | | | | | | |
> | LMDI+ | **0.9931** | **0.9310** | **0.8603** | **0.7581** | **0.9643** | **0.8521** |
> | TreeSHAP | 0.9845 | 0.7659 | 0.8080 | 0.6700 | 0.9321 | 0.7630 |
> | Local MDI | 0.9396 | 0.6042 | -0.2883 | 0.6564 | 0.8610 | 0.4863 |
>
> **Table 1.** Real Data Experiments using full datasets, excluding LIME due to runtime constraints. Bold indicates the best performance for each dataset and keep ratio.

---

> > ### Author Response · Authors · 2026-03-12
> > **Response to Reviewer LD77 [Part 2]**
> >
> > **Correlation analysis… Please add a specification of the number of signal features (and how many are correlated and not)...The paper does not explain why there is a dip in LMDI+ from 0.9 to 0.99...Since the paper stresses that LMDI+ handles very high correlations better than the other methods, there should be more values investigated between 0.85 and 0.99.**
> >
> > A: We would like to thank the reviewer for your thoughtful feedback regarding the correlation analysis in Section 4.2. We address the different components of this feedback below.
> >
> > *Regularization, Parameter Tuning, & Concavity*
> >
> > We believe the dip is driven by the interplay between the regularization strength $\alpha$
> > and the L1 ratio. Our results use the default scikit-learn `ElasticNetCV` implementation, and when inspecting the regularization selected at high $\rho$, we find that the cross-validation grid chooses values of $\alpha$ close to zero. With such weak regularization, the balance between the L1 and L2 penalties has little practical effect, and the role of the L1 ratio in shaping the results is obscured.
> >
> > To isolate this effect, we re-ran the analysis with $\alpha = 1.0$ (the scikit-learn elastic net default) while varying the `l1_ratio`, and additionally included finer-grained correlation levels between $\rho = 0.85$ and $\rho = 0.99$ as requested. Results are shown in Table 2. Under this stronger regularization regime, the L1 ratio governs the concavity of the results: with `l1_ratio` $= 0.99$ (predominantly lasso), the drop at $\rho = 0.99$ disappears entirely, whereas it persists when ridge is more prevalent. This suggests that the dip is an artifact of suboptimal regularization selection.
> >
> >
> > |Method|L1 Ratio|Feature Group|ρ=0.5|ρ=0.6|ρ=0.7|ρ=0.8|ρ=0.85|ρ=0.9|ρ=0.95|ρ=0.99|
> > |---|---|---|---|---|---|---|---|---|---|---|
> > |LMDI+|0.1|Signal|39.46|40.59|43.12|45.92|47.35|49.3|49.85|44.89|
> > |LMDI+|0.1|Cor. Non-Signal|47.81|47.89|47.93|48.6|49.04|49.02|49.67|48.97|
> > |LMDI+|0.1|Non-Signal|52.19|51.99|51.65|50.73|50.17|49.95|49.31|50.52|
> > ||||||||||||
> > |LMDI+|0.5|Signal|38.71|39.83|42.4|45.13|46.43|48.54|49.19|44.72|
> > |LMDI+|0.5|Cor. Non-Signal|48.27|48.47|48.78|49.75|50.41|50.63|51.46|50.96|
> > |LMDI+|0.5|Non-Signal|51.88|51.57|50.98|49.8|49.07|48.62|47.81|48.79|
> > ||||||||||||
> > |LMDI+|0.99|Signal|36.44|38.58|41.01|44.14|46.1|48.22|48.25|39.64|
> > |LMDI+|0.99|Cor. Non-Signal|49.08|49.5|50.61|51.94|52.2|52.52|52.9|51.04|
> > |LMDI+|0.99|Non-Signal|51.44|50.81|49.54|48.0|47.53|47.0|46.66|49.33|
> > ||||||||||||
> > ||||||||||||
> > |LIME|0.1|Signal|34.17|36.12|36.64|40.09|41.15|42.74|47.03|53.61|
> > |LIME|0.1|Cor. Non-Signal|48.97|48.79|49.47|49.52|49.67|49.74|50.22|51.25|
> > |LIME|0.1|Non-Signal|51.81|51.73|51.07|50.61|50.36|50.1|49.16|47.47|
> > ||||||||||||
> > |LIME|0.5|Signal|34.17|36.12|36.64|40.09|41.15|42.74|47.03|53.61|
> > |LIME|0.5|Cor. Non-Signal|48.97|48.79|49.47|49.52|49.67|49.74|50.22|51.25|
> > |LIME|0.5|Non-Signal|51.81|51.73|51.07|50.61|50.36|50.1|49.16|47.47|
> > ||||||||||||
> > |LIME|0.99|Signal|34.17|36.12|36.64|40.09|41.15|42.74|47.03|53.61|
> > |LIME|0.99|Cor. Non-Signal|48.97|48.79|49.47|49.52|49.67|49.74|50.22|51.25|
> > |LIME|0.99|Non-Signal|51.81|51.73|51.07|50.61|50.36|50.1|49.16|47.47|
> > ||||||||||||
> > ||||||||||||
> > |TreeSHAP|0.1|Signal|35.71|37.88|38.5|42.32|43.94|45.55|50.09|56.97|
> > |TreeSHAP|0.1|Cor. Non-Signal|48.55|48.52|49.2|49.53|49.77|50.03|50.98|52.68|
> > |TreeSHAP|0.1|Non-Signal|51.99|51.76|51.08|50.34|49.93|49.51|48.13|45.81|
> > ||||||||||||
> > |TreeSHAP|0.5|Signal|35.71|37.88|38.5|42.32|43.94|45.55|50.09|56.97|
> > |TreeSHAP|0.5|Cor. Non-Signal|48.55|48.52|49.2|49.53|49.77|50.03|50.98|52.68|
> > |TreeSHAP|0.5|Non-Signal|51.99|51.76|51.08|50.34|49.93|49.51|48.13|45.81|
> > ||||||||||||
> > |TreeSHAP|0.99|Signal|35.71|37.88|38.5|42.32|43.94|45.55|50.09|56.97|
> > |TreeSHAP|0.99|Cor. Non-Signal|48.55|48.52|49.2|49.53|49.77|50.03|50.98|52.68|
> > |TreeSHAP|0.99|Non-Signal|51.99|51.76|51.08|50.34|49.93|49.51|48.13|45.81|
> > ||||||||||||
> > ||||||||||||
> > |Local MDI|0.1|Signal|35.45|38.03|39.25|44.85|47.2|49.65|56.21|64.96|
> > |Local MDI|0.1|Cor. Non-Signal|49.04|49.25|50.37|50.82|51.54|52.45|54.08|57.91|
> > |Local MDI|0.1|Non-Signal|51.59|51.1|49.96|48.89|47.99|46.88|44.67|40.25|
> > ||||||||||||
> > |Local MDI|0.5|Signal|35.45|38.03|39.25|44.85|47.2|49.65|56.21|64.96|
> > |Local MDI|0.5|Cor. Non-Signal|49.04|49.25|50.37|50.82|51.54|52.45|54.08|57.91|
> > |Local MDI|0.5|Non-Signal|51.59|51.1|49.96|48.89|47.99|46.88|44.67|40.25|
> > ||||||||||||
> > |Local MDI|0.99|Signal|35.45|38.03|39.25|44.85|47.2|49.65|56.21|64.96|
> > |Local MDI|0.99|Cor. Non-Signal|49.04|49.25|50.37|50.82|51.54|52.45|54.08|57.91|
> > |Local MDI|0.99|Non-Signal|51.59|51.1|49.96|48.89|47.99|46.88|44.67|40.25|
> > ||||||||||||
> > ||||||||||||
> >
> > **Table 2.** Mean rank given to each feature group by various LFI methods as the amount of correlation $\rho$ is varied.

---

> > > ### Author Response · Authors · 2026-03-12
> > > **Response to Reviewer LD77 [Part 3]**
> > >
> > > *Specification of Features*
> > >
> > > We have revised the 'Setup' paragraph of Section 4.2 to explicitly specify the number of signal features, correlated non-signal features, and non-correlated non-signal features.
> > >
> > >
> > > We hope that our clarifications, revisions, and additional results provide stronger evidence for the effectiveness of LMDI+. We welcome any further suggestions you may have.

---

### Review · Reviewer_eYKg · 2026-03-07

**Summary Of Contributions:**

This paper introduces Local MDI+, a local extension of the global feature importance method MDI+.

Local MDI+ builds an expanded representation from the tree-derived basis over both in-bag and out-of-bag samples with the raw feature appended, fits a regularized GLM, and defines local feature importance via the inner product between the transformed sample representation and the fitted coefficients.


The authors argue that Local MDI+ more effectively identifies both signal and predictive features than existing methods, supported by strong results on synthetic experiments and 12 real-world benchmark datasets.

They further claim robustness under strong feature correlation, demonstrated in correlated simulations in which the true signal features remain top-ranked.

They report that Local MDI+ yields more stable local-importance rankings across random seeds and selects the fewest unique features across repeated fits.

They also claim the utility of Local MDI+ in real-world application settings from the perspectives of counterfactual explanations and subgroup discovery.

**Audience:**

Yes

**Audience Explanation:**

Although this paper focuses on simple ML tasks, at least some of TMLR's audience would be interested in the findings, as the proposed method could be useful in real-world applications.

**Broader Impact Concerns:**

From my perspective, I do not see any clear ethical implications.

**Claims And Evidence:**

Yes

**Claims Explanation:**

This paper clearly explains the proposed method, LMDI+, which comprises three key modules: OOB samples, feature transformations, and GLMs.

The experiments provide strong support for the effectiveness of the proposed method at accurately classifying "signal features" and "predictive features" in both synthetic and real datasets, as well as its robustness, stability, and usability in appropriately controlled experimental settings, including experiments under the strong-correlation setting, under repeated random seed initialization, and in classification benchmarks.

**Requested Changes:**

- I read the main contribution primarily as extending MDI+ to the sample-specific setting and demonstrating empirical gains in local feature identification, stability, and selected downstream uses, rather than as a broad introduction of local tree-based explanations in general. In that context, the paper may benefit from discussing MAPLE [Plumb+, NeurIPS-2018], which combines random forests with local linear explanations and also uses them to reveal structure beyond per-instance explanations. This seems particularly relevant to the paper's discussion of subgroup discovery.
  - [Plumb+, NeurIPS-2018] Model Agnostic Supervised Local Explanations

- For the stability and trustworthiness claims, one relevant missing citation could be [Burger+, EMNLP-2023], which studies the stability of LIME explanations in text settings and shows that explanations can be manipulated substantially while preserving semantics and the original prediction. Although this is not about tabular tree models, it supports the broader point that stability matters in high-stakes settings. Citing it could also help clarify that the paper's contribution is not to introduce stability as a concern, but to demonstrate improved stability for the proposed method under the paper's evaluation protocol.
  - [Burger+, EMNLP-2023] Are Your Explanations Reliable? Investigating the Stability of LIME in Explaining Text Classifiers by Marrying XAI and Adversarial Attack

- The feature-identification claims may also benefit from acknowledging recent work on explanation quality and evaluation. *Selective Explanations* [Paes+, NeurIPS-2024] argues that feature attributions can be misleading and proposes a way to detect and improve low-quality explanations. Relatedly, [Edin+, ACL-2025] show that AOPC can produce misleading cross-model comparisons and propose a normalized alternative. Even if AOPC is not used here, this line of work suggests that empirical claims about “better explanations” can depend on the evaluation setup and may therefore benefit from slightly more careful framing.
  - [Paes+, NeurIPS-2024] Selective Explanations
  - [Edin+, ACL-2025] Normalized AOPC: Fixing Misleading Faithfulness Metrics for Feature Attributions Explainability

- The phrase "exact linear equivalence" in Section 3.3 may read somewhat more strongly than what Section 3.2 explicitly establishes, since Section 3.2 presents the equivalence for OLS on the transformed basis. In contrast, Section 3.3 introduces an augmented representation together with a regularized GLM.

- Section 3.3 defines the score for feature $k$ as $ \tilde{\Psi}^{(k)}(x)^\top \hat{\beta}^{(k)}_\lambda $, but it would be helpful to clarify how this quantity should be interpreted when a GLM link function is used. In particular, it is not entirely clear whether the score should be viewed as a contribution to the predicted probability, the linear predictor, or some other quantity.

- The paper explains how feature scores are computed and ranked, but it does not explicitly state what $\sum_k \mathrm{LMDI}^+_k(x)$ corresponds to. It may therefore be helpful to clarify whether these scores are intended to sum to the prediction, the linear predictor, or primarily to serve as ranking scores.

- The rule that sets the score to zero when $S^{(k)}=\emptyset$ would also benefit from a bit more explanation. Since Section 3.3 defines an augmented representation for each feature by appending $x_k$, it is not immediately obvious why a feature that never appears in a split should automatically receive a zero local score. This may well be intentional, but the rationale could be made clearer.

- Section 4.3 evaluates selected features by retraining a model on them and measuring test $R^2$/AUROC. This is a reasonable proxy for predictive usefulness, but it does not directly validate the correctness of the local feature rankings themselves. The claim may therefore be best framed in terms of the retraining-based evaluation used in the paper.

- Section 5 measures stability using the number of unique top-k features across repeated fits. This captures top-feature-set consistency, but not necessarily the stability of the full ranking or attribution magnitudes. The paper may therefore want to frame this more narrowly as top-k selection stability.

- The robustness/generalization claim in Section 6 may read somewhat strongly. The analysis varies only a limited subset of the RF hyperparameters and adds one gradient-boosting setting. This is useful evidence, but still somewhat limited. Claims about robustness to ensemble structure or broader generalization beyond RFs may therefore be best restricted to the specific settings tested.

- Section 7 shows that LMDI+ retrieves closer counterfactuals within the paper's LFI-based matching procedure, measured by $ \ell_1 $ distance. That does not directly establish broader notions such as feasibility, plausibility, or respect for immutable features. The evidence seems to support "closer counterfactuals," whereas "more actionable interventions" may read somewhat too strongly.

- Section 8 evaluates subgroup discovery through a single case study on Miami Housing. This is an interesting illustration, but the evidence is still limited for a broad subgroup-discovery claim. The paper may be stronger if this is framed more explicitly as a case study or downstream use case rather than a general strength of the method.

---

> ### Author Response · Authors · 2026-03-12
> **Response to Reviewer eYKg [Part 1]**
>
> We thank the reviewer for their thoughtful feedback. We have addressed your concerns below and revised the paper accordingly. The updated sections are highlighted in blue.
>
> **Requested missing citations**
>
> A: We thank the reviewer for bringing up these relevant papers and have incorporated them accordingly. [Plumb+, NeurIPS-2018] has been added to the Related Work section with a description. [Burger+, EMNLP-2023] has been added in Section5 alongside the discussion of stability. [Paes+, NeurIPS-2024] and [Edin+, ACL-2025] have been added in Section 4.3 under real-world feature selection experiments.
>
> **The phrase "exact linear equivalence" in Section 3.3 may read somewhat more strongly than what Section 3.2 explicitly establishes, since Section 3.2 presents the equivalence for OLS on the transformed basis. In contrast, Section 3.3 introduces an augmented representation together with a regularized GLM.**
>
> A: Thank you for pointing this out. The phrase "exact linear equivalence" in Section 3.3 refers to a different notion than the equivalence established in Section 3.2. Section 3.2 shows that a fitted decision tree is equivalent to an OLS model on the transformed basis, while Section 3.3 describes how the Local MDI+ score for the $k$th feature is the product $\tilde{\Psi}^{(k)}\left(\mathbf{x}\right)^\top\hat{\boldsymbol{\beta}}^{(k)}\_{\lambda}$, which is linear in the $\hat{\boldsymbol{\beta}}$s. We agree the original phrasing was ambiguous and have revised Section 3.3 to distinguish these two notions of linearity more clearly.
>
> **It would be helpful to clarify how the score for feature $k$ (Section 3.3) should be interpreted when a GLM link function is used. In particular, it is not entirely clear whether the score should be viewed as a contribution to the predicted probability, the linear predictor, or some other quantity.**
>
> A: The choice of GLM affects how the coefficients $\hat{\beta}^{(k)}\_{\lambda}$ are estimated. For instance, logistic regression is used for classification and linear regression for regression tasks. However, regardless of the GLM used, the score $\tilde{\Psi}^{(k)}(x)^\top \hat{\beta}^{(k)}\_{\lambda}$ represents the contribution of feature $k$ to the linear predictor: for regression this corresponds to a contribution to the predicted output, and for classification it corresponds to a contribution to the log-odds. This preserves the additivity of the linear predictor across features and provides a consistent basis for ranking features across different settings.
>
> **It may therefore be helpful to clarify whether these scores are intended to sum to the prediction, the linear predictor, or primarily to serve as ranking scores.**
>
> A: The scores are primarily intended as ranking scores rather than quantities that sum to the prediction. We note that this is by design, as LMDI+ modifies the underlying RF framework to produce feature importance scores that better reflect the underlying DGP rather than to reconstruct the model output.
>
> **Since Section 3.3 defines an augmented representation for each feature by appending $x_k$, it is not immediately obvious why a feature that never appears in a split should automatically receive a zero local score. This may well be intentional, but the rationale could be made clearer.**
>
> A: Thank you for raising this. The zero score is intentional: If a feature never appears in any split of a given tree, it contributes no structural information to that tree's predictions, and its score is therefore zero by definition. We have revised Section 3.3 to make this explicit.
>
> **Section 4.3 evaluates selected features by retraining a model on them and measuring test $R^2$/AUROC. This is a reasonable proxy for predictive usefulness, but it does not directly validate the correctness of the local feature rankings themselves.**
>
> A: Directly validating the correctness of local feature rankings on real-world data is unfortunately impossible, as ground truth is unavailable. We therefore follow the commonly used remove-and-retrain evaluation technique [1, 2]. The correctness of local feature rankings is instead evaluated in Sections 4.1 and 4.2, where synthetic settings allow us to compare against known ground-truth features.
>
> [1] Sara Hooker, Dumitru Erhan, Pieter-Jan Kindermans, and Been Kim. A benchmark for interpretability methods in deep neural networks. Advances in neural information processing systems, 32, 2019.
>
> [2] Nourah Alangari, Mohamed El Bachir Menai, Hassan Mathkour, and Ibrahim Almosallam. Exploring evaluation methods for interpretable machine learning: A survey. Information, 14(8):469, 2023.

---

> > ### Author Response · Authors · 2026-03-12
> > **Response to Reviewer eYKg [Part 2]**
> >
> > **Section 5 measures stability using the number of unique top-k features across repeated fits. This captures top-feature-set consistency, but not necessarily the stability of the full ranking or attribution magnitudes.**
> >
> > A: We acknowledge that the unique top-k feature metric captures consistency of the selected top feature set rather than the full ranking. However, we note that top-k set consistency is the most practically relevant notion of stability, as downstream use cases such as feature selection are driven by the identity of the top features rather than the full ranking or precise magnitudes.
> >
> > To further validate our stability claims, we have conducted an additional experiment measuring the standard deviation of feature selection performance (Section 4.3) across repeated fits with different random seeds. As shown in the following table, LMDI+ consistently achieves lower standard deviation than all baselines, indicating that its improved set-level stability is accompanied by more stable predictive performance across runs.
> >
> > | Method | Keep 10% | Keep 20% | Keep 30% | Keep 40% |
> > |---|---:|---:|---:|---:|
> > | LMDI+ | **0.0115** | **0.0082** | **0.0050** | **0.0046** |
> > | LIME | 0.0220 | 0.0175 | 0.0133 | 0.0096 |
> > | TreeSHAP | 0.0125 | 0.0094 | 0.0058 | 0.0048 |
> > | Local MDI | 0.0163 | 0.0103 | 0.0063 | 0.0058 |
> >
> > **Table 1.**  Standard deviation of feature-selection performance across five random seeds, averaged across datasets and reported for varying top-feature keep ratios per sample. Bold indicates the lowest standard deviation for each keep ratio. LMDI+ consistently achieves the lowest standard deviation across keep ratios, indicating more stable performance across runs.
> >
> > **The robustness/generalization claim in Section 6 may read somewhat strongly...Claims about robustness to ensemble structure or broader generalization beyond RFs may therefore be best restricted to the specific settings tested.**
> >
> > A: We thank the reviewer for this observation. We have expanded the Discussion section to clarify this limitation of Section 6, and note that a more exhaustive evaluation of robustness across ensemble structures remains an important direction for future work.
> >
> > **Section 7 shows that LMDI+ retrieves closer counterfactuals within the paper's LFI-based matching procedure… the evidence seems to support "closer counterfactuals," whereas "more actionable interventions" may read somewhat too strongly.**
> >
> > A: We agree that there is a subtle difference between “close counterfactuals” and “actionable interventions”, as not all features are able to be intervened on. We have revised this sentence in Section 7 to emphasize closer counterfactuals instead.
> >
> > **Section 8 evaluates subgroup discovery through a single case study on Miami Housing…The paper may be stronger if this is framed more explicitly as a case study or downstream use case rather than a general strength of the method.**
> >
> > A: Thank you for pointing this out. We have revised the abstract, introduction, and the title of Section 8 to explicitly frame this as a case study.
> >
> > We hope that our clarifications and revisions provide stronger evidence for the effectiveness of LMDI+. We welcome any further suggestions you may have.

---

### Decision · Action_Editor_ULMy · 2026-04-29

**Recommendation:** Accept as is

**Audience:**

Yes

**Audience Explanation:**

Tree-based methods are commonly used models and the proposed method for instance-level explanations could be a valuable tool for identifying biases or mistakes. The manuscript should be of interest to many in the TMLR audience.

**Claims And Evidence:**

Yes

**Claims Explanation:**

The manuscript proposes a local version of MDI+, a global feature importance method for tree-based models. The performance claims were supported by experimental evidence. However, one reviewer mentioned that the datasets were all subsampled to at most 2,000 samples. The author’s rebuttal explained that this was to allow fair comparisons with LIME for which runs with larger datasets are computationally prohibitive. The rebuttal also provided the results on the full benchmarks, excluding LIME, which showed that the proposed method still outperforms TreeSHAP and Local MDI across benchmarks and feature retention levels. After the rebuttal, reviewers all recommended acceptance of the revised manuscript, which also incorporated suggested changes on related work, important clarifications, and changes in the structure of the paper.